

**Impacts of aerosol-radiation interaction on meteorological forecast**
**over northern China by offline coupling the WRF-Chem simulated**
**AOD into WRF: a case study during a heavy pollution event**
Yang Yang[1], Min Chen[1], Xiujuan Zhao[1*], Dan Chen[1*], Shuiyong Fan[1],
and Shaukat Ali[2]
*1 Institute of Urban Meteorology, China Meteorological Administration, Beijing*
*100089, China*
*2 Global Change Impact Studies Centre, Ministry of Climate Change, Islamabad*
*44000, Pakistan*



**Abstract**

To facilitate the future inclusion of aerosol-radiation interactions in the regional

operational Numerical Weather Prediction (NWP) system – RMAPS-ST (adapted

from Weather Research and Forecasting, WRF) at the Institute of Urban

Meteorology (IUM), China Meteorological Administration (CMA), the impacts of

aerosol-radiation interactions on the forecast of surface radiation and meteorological

parameters during a heavy pollution event (December $6^{th}$ -$10^{th}$, 2015) over northern

China were investigated. The aerosol information was simulated by RMAPS-Chem

(adapted from WRF model coupled with Chemistry, WRF-Chem) and then

offline-coupled into Rapid Radiative Transfer Model for General Circulation Models

(RRTMG) radiation scheme of WRF to enable the aerosol-radiation feedback in the

forecast. To ensure the accuracy of high-frequent (hourly) updated aerosol optical

depth (AOD) field, the temporal variations of simulated AOD at 550nm were

evaluated against satellite and in-situ observations, which showed great consistency.

Further comparison of $PM_{2.5}$ with in-situ observation showed WRF-Chem

reasonably captured the $PM_{2.5}$ field in terms of spatial distribution and magnitude,

with the correlation coefficients of 0.85, 0.89 and 0.76 at Beijing, Shijiazhuang and

Tianjin, respectively. Forecasts with/without the hourly aerosol information were

conducted further, and the differences of surface radiation, energy budget, and

meteorological parameters were evaluated against surface and sounding

observations. The offline-coupling simulation (with aerosol-radiation interaction



active) showed a remarkable decrease of downward shortwave (SW) radiation
reaching surface, thus helping to reduce the overestimated SW radiation during
daytime. The simulated surface radiation budget was also improved, with the biases
of net surface radiation decreased by 85.3%, 50.0%, 35.4%, and 44.1% during
daytime at Beijing, Tianjin, Taiyuan and Jinan respectively, accompanied by the
reduction of sensible (16.1 W m$^{-2}$, 18.5%) and latent (6.8 W m$^{-2}$, 13.4%) heat fluxes
emitted by the surface at noon-time. In addition, the cooling of 2-m temperature
(~0.40 °C) and the decrease of horizontal wind speed near surface (~0.08 m s$^{-1}$)
caused by the aerosol-radiation interaction over northern China helped to reduce the
bias by ~73.9% and ~7.8% respectively, particularly during daytime. Further
comparisons indicated that the simulation implemented AOD could better capture
the vertical structure of atmospheric wind. Accompanied with the lower planetary
boundary layer and the increased atmospheric stability, both U and V wind at
850hPa showed the convergence which were unfavorable for pollutants dispersion.
Since RMPAS-ST provides meteorological initial condition for RMPS-Chem, the
changes of meteorology introduced by aerosol-radiation interaction would routinely
impact the simulations of pollutants. These results demonstrated the profound
influence of aerosol-radiation interactions on the improvement of predictive
accuracy and the potential prospects to offline couple near-real-time aerosol
information in regional RMAPS-ST NWP in northern China.
**Key words:** Aerosol-radiation interactions, offline-coupling, WRF, northern China,





pollution



## 1. Introduction

Aerosol-radiation interactions modify the radiative energy budget of the
earth-atmosphere system through the interaction between aerosols and solar radiation
by scattering and absorbing mechanism as well as the absorption and emitting of
thermal radiation (Ramanathan et al., 2001; Yu et al., 2006). The aerosol-radiation
interaction may cool or heat the earth-atmosphere system, alter surface and
atmospheric radiation and temperature structure on regional and global climate, which
have been widely reported and studied (Hansen et al., 1997; Ramanathan et al., 2001;
Kaufman et al., 2002; Liao et al., 2006; Zhang et al., 2010; Ghan et al., 2012; Yang et
al., 2017a). Considering the lifetime of most aerosol particles and their locally uneven
distribution, as well as their high dependence on emission sources and local
meteorological conditions for dispersion (Rodwell and Jung, 2008; Liu et al., 2012;
Liao et al, 2015), the impacts of aerosol in short durations over regional areas are
worthy of more concerns (Cheng et al., 2017; Zheng et al., 2019).
With substantial aerosol loading, aerosol particles have significant influences on
meteorology, and many endeavors by both field experiments and numerical models
have been devoted to study the impacts of aerosol-radiation interaction on
meteorological fields, including surface solar radiation, planetary boundary layer
(PBL), atmospheric heating rate, atmospheric stability (Hansen et al., 1997; Ackerman
et al., 2000; Quan et al., 2014; Yang et al., 2017b; Wang et al., 2018), cloud formation
due to thermodynamic changes, and further the onset or reduction of precipitation



systems (Grell et al., 2011; Guo et al., 2016). For instance, in worldwide, the
simulations with Weather Research and Forecasting (WRF) model coupled with
Chemistry (WRF-Chem) showed that by purely taking into account the
aerosol-radiation interactions, aerosols may reduce incoming solar radiation by up to
−9% (−16%) and 2-m temperatures by up to 0.16°C (0.37°C) in January (July) over
the continental U.S. (Zhang et al., 2010) , affect meso-scale convection system owing
to thermodynamic changes over Atlantic Ocean during Saharan dust eruption period
(Chen et al., 2017), and lead to the distinct changes in precipitation due to the changes
in temperature profile and stabilities induced by the aerosol-radiation interaction over
Eastern China (Huang et al., 2016).

Northern China is experiencing heavy air pollution in past two decades, with

particle matter (PM) being the primary pollutant, particularly during wintertime (Chan
and Yao, 2008; Zhang et al., 2015; Zhao et al., 2019) due to the combination of high
primary and precursor emissions and frequent stable meteorological conditions in this
area (Elser et al., 2016; Zhang et al, 2018). The effects of aerosol-radiation interaction
on meteorology were expected to be much more significant over northern China.
Applying WRF and Community Multi-scale Air Quality Model (CMAQ) system
(WRF-CMAQ), Wang et al. (2014) and Sekiguchi et al. (2018) reported a 53%
reduction in solar radiation reaching surface and ~100m decrease of planetary
boundary layer height (PBLH) in response to the presence of aerosols during a severe
winter haze episode in China. Wang et al. (2015a, b) used the online chemical weather



forecasting mode Global/Regional Assimilation and PrEdiction System/ Chinese
Unified Atmospheric Chemistry Environment (GRAPES/CUACE) and illustrated that
the solar radiation at ground decreased by 15% in Beijing–TianJin–Hebei, China, and
its near surroundings, accompanied by the decrease in turbulence diffusion of about
52% and a decrease in PBLH of about 33 % during a haze episode of summertime in

2008.

Considering the significant influence of the aerosol-radiation interaction on

meteorological forecasts as illustrated in many studies (Kaufman et al., 2002; Zhang
et al., 2010), several weather forecast centers are conducting research to facilitate
more complex aerosol information inclusion in operational numerical weather
prediction (NWP) models. For example, Rodwell and Jung (2008) showed the local
medium-range forecast skills were improved due to the application of new
climatological aerosol distribution in European Centre for Medium-Range Weather
Forecasts (ECMWF). Recently, a positive impact up to a 48h lead time on the 2m
temperature and forecasts of surface radiative fluxes were reported in ECMWF by
applying the prognostic aerosols compared to the monthly climatological aerosol
(Rémy et al., 2015). Toll et al. (2016) found that the inclusion of aerosol effects in
NWP system was beneficial to the accuracy of simulated radiative fluxes, temperature
and humidity in the lower troposphere over Europe. In addition, it was shown that the
quality of weather forecasts at UK MET office can be further advanced when the
real-time aerosol distribution rather than climatological distribution were included,





with the decreased bias of downward SW at surface ($-2.79$ W m$^{-2}$ vs. $-5.30$ W m$^{-2}$)
and the mean sea-level pressure (0.71hPa vs. 0.80hPa) (Mulcahy et al., 2014; Toll et
al., 2015). For these research serving for operational NWP systems, offline approach
(that aerosol information were simulated by separate chemistry system and then
offline coupled to NWP model) were mostly used.

In most previous research-targeted modeling studies over northern China, the

aerosol-radiation interaction has been widely accessed in online-coupled
meteorology-chemistry models, which might not be practical for NWP purpose.
Considering aerosol particles differ by morphology, size and chemical composition,
therefore, the numerical treatment of aerosol particles in atmospheric models needs
sophisticated method and considerable simplifications, which may bring in more
assumptions and uncertainties in online coupling (Baklanov et al., 2014). Moreover,
the online simulations require quite high computational costs and could not meet the
requirement of efficiency for operational NWP. Grell and Baklanov (2011) illustrated
that the offline approach could generate to almost identical results compared to online
simulation with the offline-coupling intervals about 0.5-1h. Thus, the
computational-economic offline simulation provides a feasible and computationally
less demanding approach to include the aerosol-radiation interaction in an operational
NWP system. Péré et al. (2011) adopted an offline-coupling between the
chemistry-transport model CHIMERE and WRF to study the radiative forcing of high
load aerosols during the heat wave of summer in 2003 over Western Europe. Wang et



al. (2018) offline implemented the daily AOD from Moderate Resolution Imaging
Spectroradiometer (MODIS) to WRF during a heavy winter pollution at Beijing to
study the effect of aerosols on boundary layer. Still, there have been few studies that
adopted offline simulation to investigate the impacts of aerosol-radiation interactions
over northern China on NWP system. At Institute of Urban Meteorology, regional
operational NWP system–RMAPS-ST (adapted from WRF) and regional air quality
model–RMPSA-Chem (adapted from WRF-Chem) were applied operationally. In this
study, we investigate the radiative effects of aerosols and their feedbacks on weather
forecasting over northern China during a polluted event occurred in winter of 2015,
and further potential impacts of changed meteorology to the transport and dissipation
of pollution. The simulations were in the configurations of the two systems, aiming at
presenting the offline-coupling of the high-frequent real-time aerosol distribution
simulated by WRF-Chem and WRF, and evaluating the potential effects of
aerosol-radiation interactions on the forecast skills in the RMAPS-ST system for
future purpose.

The remainder of the paper was organized as follows. Section 2 presented the

model configuration and experimental design. In section 3, the model's capabilities in
capturing and forecasting the pollution episode were validated with observations first,
and impacts of aerosol-radiation interactions on meteorological forecasting over
northern China were analyzed further. The final section provided the concluding
remarks.



## 2. Model description and experimental design


WRF is a state-of-the-art atmospheric modeling system designed for both
meteorological research and NWP. The WRF version 3.8.1 released in August, 2016
was used in this study for a domain covering the northern China with a horizontal
resolution of 9km (222×201 grid points, Fig. 1a), and for 50 vertical levels. The
lateral boundary condications (BCs) and initial conditions (ICs) for meteorological
variables are provided by the forecast of ECMWF. The major physical schemes
include the Assymetric Convective Model Version 2 (ACM2) PBL scheme (Pleim,
2007), the Thompson microphysics without aerosol-aware option (Thompson et al.,
2008), the Kain-Fritsch cumulus parameterization (Kain, 2004), and the Natioal
Center for Envirometal Prediction, Oregon State University, Air Force, and
Hydrologic Research Lab's (NOAH) land-surface module (Chen and Dudhia, 2001;
Ek et al., 2003). The landuse data have been reprocessed, which has a higher
accuracy and finer classification for urban areas (Zhang et al., 2013) and the urban
canopy model (UCM) was not actived.
The shortwave and longwave radiation scheme is Rapid Radiative Transfer
Model for General Circulation Models (RRTMG) (Iacono et al., 2008). RRTMG
scheme is a new version of RRTM added in Version 3.1, and includes the Monte
Carlo Independent Column Approximation (MCICA) method of random cloud
overlap. A recent intercomparison study showed that RRTMG had relativlely smaller
mean errors in solar flux at the surface and the top of the atmosphere (Oreopoulos et



al., 2012) and was considered as recommended WRF configuration for air quality
modeling (Rogers et al., 2013). RRTMG scheme is capable to include the
climatological aerosol data with spatial and temporal variations or an external time
varing 3D aerosol input through the option of AER_OPT (Ruiz-Arias et al., 2014).
In the present study, the real-time hourly aerosol optical depth (AOD) at 550nm
from external files were input into WRF following the second approach. The AOD
at 550nm was calculated as the vertical intergration of extinction coefficients at
550nm from WRF-Chem simulation.

WRF-Chem version 3.3.1 was applied in this study, and the horizontal

resolution was 9 km, with $222 \times 201$ grid points covering northern China, which were
the same as configurions of WRF mentioned above. WRF-Chem simulates the
formation, transformation and transport processes of both primary and secondary
atmospheric pollutants, including gases and PM species (Zhao et al., 2019). Physical
parameterizations included single-layer Urban Canopy Model, Noah land-surface,
Yonsei University (YSU) PBL, Grell-Devenyi ensemble convection, Thompson
microphysics, and RRTM longwave and Goddard shortwave radiation (Chen and
Dudhia, 2001; Hong et al., 2006; Grell and Dévényi, 2002; Thompson et al., 2008;
Mlawer et al., 1997; Chou and Suarez, 1999). Carbon bond mechanism Z (CBMZ)
including comprehensive reactions and alterable scenarios were used as the
gas-phase mechanism. Model for Simulating Aerosol Interactions and Chemistry
(MOSAIC) are used with four size bins (Zaveri and Peters, 1999). Anthropogenic


emission data were from the MEIC (2012) inventory (http://www.meicmodel.org/)
with a resolution of $0.1°\times0.1°$. Meteorological ICs and BCs were obtained from the
Final Analysis data (FNL) with a resolution of $1.0°\times1.0°$ from the National Centers
for Environmental Prediction (NCEP). To generate aerosol fields for study period
(Dec. $2^{nd}$-$11^{th}$), 9-days WRF-Chem simulations from Dec. $2^{nd}$ were conducted using
prescribed idealized profiles as ICs and BCs for chemical species.
To estimate the aerosol radiative forcing and its feedbacks on meteorological
fields, two sets of 24-hour WRF forecasts were conducted at 00UTC from $2^{nd}$-$10^{th}$
December 2015 with WRF-Chem simulated AOD fields as input fields. The only
difference between the two sets of forecasts is whether the aerosol radiative
feedback is activated (Aero) or not (NoAero), and other schemes remained the same.
The sites of observations over simulated domain and northern China plain (NCP,
purple box in Fig. 1a) are shown in Fig. 1. Since the AOD provided by MODIS
instruments on-board NASA polar orbiting satellites Aqua and Terra are both not
available in the region with high pollution, three sites of AErosol Robotic NETwork
(AERONET) are used to validate the simulation (black dots in Fig. 1b), and the
observed AOD obtained from observation at the Institute of Atmospheric Physics
(IAP), Chinese Academy of Sciences (39°58′ 28″ N, 116°22′ 16″ E) in Beijing
city (blue dot in Fig. 1b) is also included as supplementary. The hourly observed
$PM_{2.5}$ concentrations of total 813/332 monitoring stations over the study
domain/NCP were from the released data by the China National Environmental





Monitoring Centre (http://106.37.208.233:20035/, colored dots in Fig. 3a). For given
cities (dots in Fig. 1a), hourly $PM_{2.5}$ concentration was represented by the average of
data from all monitoring sites located in the city. Simulated meteorological variables
including 2-m temperature and wind speed at 10m were evaluated using in-situ
observations from National Meteorological Information Center
(http://data.cma.cn/data/cdcindex.html) of China Meteorological Administration
(CMA, dots in Fig. 8a). The radiations were observed at IAP and in-situ stations of
CMA (shown as triangles in Fig. 1a). The vertical observation of atmospheric wind
speed from sounding were also used (circles in Fig. 1a). The variables, sources,
numbers of sites in the domain and NCP and the frequency of chemical and
meteorological observations were also listed in Table 1.
**3. Results**
**3.1 Evaluation of AOD and $PM_{2.5}$ simulated by WRF-Chem**

Before the offline-coupling of the WRF-Chem simulated hourly AOD to

meteorological model WRF, we first validated the simulated AOD and ensured the
model's capability to reproduce the features of the aerosol field. Figure 2 displayed
the temporal variation of simulated AOD at 550nm (blue solid) at four sites, in
comparison with three AERONET stations (black circles in Figs. 2a-c) and IAP site
(black circles in Fig. 2d) for the period during 3rd to 11th Dec, 2015 (local time, LT).
As shown in blue solids in Fig. 2a, the simulated AOD increased since 6th Dec. and
reached the peak value of 9 on 7th, and the high AOD value maintained until 9th and



reached the second peak. The second peak was also observed from AERONET
though most of them were missing during the pollution event. The temporal
variations of AOD at Beijing-CMA and IAP (Figs. 2b and d) were analogical with
those at Beijing station (Fig. 2a). Meanwhile, the simulated AOD at Xianghe (Fig.
2c) was relatively lower than those at other stations; it might be that Xianghe is a
rural station and was less polluted than urban station during this episode.

Considering that the available observational AOD data was quite limited, and the

aerosol extinction was mainly attributed to scattering and absorption of solar
radiation by $PM_{2.5}$ and their hygroscopic growth with relative humidity (Cheng et al.,
2006), next we compared the simulated $PM_{2.5}$ concentrations with corresponding
in-situ observation over the model domain. As shown in Fig. 3, the simulated and
observed pollution were both initiated over Henan province on $6^{th}$, further
intensified and shifted northward afterwards. The polluted center located over south
of Hebei province and maintained until $10^{th}$, with the maximum $PM_{2.5}$ concentration
exceeding $440\mu g\ m^{-3}$. The results indicated that WRF-Chem could well capture the
spatial features of $PM_{2.5}$ and its temporal variation, in spite of the slight discrepancy
of the center position during $9^{th}$ and $10^{th}$.

To further assess the temporal evolutions of the pollution, the simulated $PM_{2.5}$

concentrations at three major cities (Beijing, Shijiazhuang and Tianjin, shown as
black dots in Fig. 1a) in northern China were compared with those observation as
shown in Fig. 4. It showed that the hourly variations of $PM_{2.5}$ concentration



including the occurrence of several high peaks at the three cities could be reasonably
reproduced by WRF-Chem, despite the slight overestimation (underestimation) of
the peak magnitude during 9[th] to 10[th] at Beijing and Shijiazhuang (Tianjin). The
correlation coefficients (R) between simulation and observation at Beijing,
Shijiazhuang and Tianjin were 0.85, 0.89 and 0.76, respectively.
**3.2 Aerosol effects on meteorological simulations**
In this section, the influences of aerosol-radiation interaction on the spatial and
temporal variations of radiation and energy budget simulated by WRF model were
analyzed, and their impacts on the forecasts of meteorological fields were discussed
further.
**3.2.1   Aerosol impacts on simulations of radiative forcing and heat fluxes**
To illustrate the impacts of aerosol-radiation interaction on the forecasts of
radiation during the pollution event, the simulated surface downward SW radiation
and net radiation at Beijing, Tianjin, Taiyuan and Jinan, as denoted by the triangles
in Fig. 1a, were compared with observations in Fig. 5. To show the relationship with
aerosol, the time series of AOD for Dec. 3[th] -11[th] were overlay as gray shadings in
Fig. 5. During the clean stage with quite low AOD values (close to 0) before 6[th] Dec.,
both simulations with and without aerosols reasonably reproduced the temporal
variation of downward SW at Beijing despite the slightly overestimation during the
noon-time (Fig. 5a). However, the overestimated downward SW in NoAero turned
to intensify extensively since 6[th] Dec. and sustained till 10[th] Dec., accompanied by



the occurrence of the pollution with the high AOD value. Meanwhile, the downward
SW was much lower in Aero than that in NoAero due to aerosol extinction, with
resembled temporal variations and comparable magnitude at the peak time compared
to the observations. Similarly, the variations of downward SW from Aero simulation
were also closer to observations at Tianjin, Taiyuan and Jinan than those in NoAero
(Figs. 5b-d). It was noted that the most significant improvement of simulated
downward SW at Jinan appeared on $10^{th}$ Dec. and was later than that at Beijing,
which was consistent with the AOD's variations at Jinan. Moreover, the surface
energy balance was also affected by the reduction of downward SW radiation
reaching the ground due to the presence of aerosol particles. As shown in Figs. 5e–h,
in corresponding to the changes in downward SW, the variations of net radiation at
surface in Aero were also in better agreement with observation during the polluted
period than in NoAero, particularly during daytime with the high AOD values.
To further quantify the influence of the aerosol-radiation interaction on the
diurnal variation of surface radiation, next we compared the simulated averaged
diurnal variation of downward SW and net radiation during the polluted episode ($6^{th}$
to $10^{th}$) with observation. Figure 6a showed that there existed a large overestimation
of surface downward SW during the daytime in NoAero. Particularly, the
overestimated downward SW tented to increase since morning (0800 LT) and peak
at noon (1300 LT) with the maximum bias reaching 226.5 W m$^{-2}$, and the mean bias
of ~149.4 W m$^{-2}$ during daytime (averaged during 0800 to 1800 LT, Table 2).



However, the overestimated SW radiation was remarkably reduced in Aero with the
mean bias of 38.0 W m$^{-2}$ during daytime. Similarly, the diurnal variation and
magnitude of downward SW radiation at surface were also better captured at Tianjin,
Taiyuan and Jinan in Aero (Figs. 6b–d), with the lower bias (70.9 W m$^{-2}$, 118.3 W
m$^{-2}$ and 97.7 W m$^{-2}$) than in NoAero (115.5 W m$^{-2}$, 155.0 W m$^{-2}$ and 149.1 W m$^{-2}$)
during daytime. Consistent with this finding, the reduction of downward SW was
also reported in United States (Zhang et al., 2010) and Europe (Toll et al., 2016)
with relatively lower decrease (10 W m$^{-2}$ and 18 W m$^{-2}$); the relatively larger
reductions (30-110 W m$^{-2}$) in northern China is possibly due to the higher aerosol
load. Figures 6e–h presented the diurnal variations of net radiation, with positive
(negative) net radiation during daytime (nighttime) in observation, and the NoAero
tended to overestimate (underestimate) the net radiation at surface during daytime
(nighttime), indicating that there existed surplus energy income and outcome in
model than those in observation, inducing the larger magnitude of diurnal cycle of
net radiation. By including the aerosol-radiation interaction in the model, the
simulated diurnal variations of net radiation were markedly improved, particularly
during daytime with the reduction of bias by 85.3%, 50.0%, 35.4%, and 44.1% at
Beijing, Tianjin, Taiyuan and Jinan, respectively.

In response to the decrease of downward SW radiation and net radiation at the

ground during daytime, the surface fluxes also changed in presence of aerosol
extinction within the energy-balanced system. Figure 7 displayed the difference of





surface sensible and latent heat flux between Aero and NoAero at 1300LT, when the
influences of the aerosol on radiation reaching the peak. Comparing to the NoAero
simulation, both the surface sensible and latent heat flux emitted by the surface were
reduced in the Aero simulation, with the domain-average of 16.1 W m$^{-2}$ (18.5%) and
6.8 W m$^{-2}$ (13.4%) respectively. It was noted that the decrease of the surface latent
heat flux was less pronounced than that of surface sensible heat flux, suggesting the
impact of aerosol-radiation interaction on the humidity was less significant than that
of temperature, which was also reported over United States (Fan et al., 2008) and
western Europe (Péré et al., 2011).
**3.2.2   Aerosol impacts on simulations of temperature, PBLH and wind fields**
The changes in radiation and energy budget through the impacts of
aerosol-radiation interaction would certainly induce the changes in PBL
thermodynamics and dynamics, which would result in changes in the forecasts of
meteorological fields. The impacts on the forecasts of 2-m temperature, PBLH and
wind fields due to the aerosol-radiation interaction were discussed in the following
subsection.
Figure 8 presented the diurnal variation of averaged bias of 2-m temperature
during polluted period in NoAero (upper panel) and Aero (lower panel) compared
with the in-situ observation during 1100 LT to 2300 LT. It was obvious that the
temperature of NoAero was significantly overestimated for a wide range over
northern China, particularly over the plain areas including south of Hebei, Henan



and Shanxi provinces. The warm biases tended to intensify in the afternoon and
reach ~3°C over south part of Hebei province (Figs. 8b–c). Accompanied by the
warm biases over plain areas throughout the day, the mountain areas were
dominated by the cold biases until 1700 LT, and turned to be warm biases afterwards,
which were attributed by the frozen water in soil due to wet bias of soil moisture
over mountain areas, inducing overestimated energy transport from atmosphere to
soil during daytime. Compared to NoAero, the lower temperature in Aero due to the
decreased surface solar radiation, caused by aerosol extinction leaded to the reduced
warm bias in NCP region. However, the cold bias in Beijing area was slightly
intensified, which may partly relevant with the overestimated $PM_{2.5}$ concentration in
Beijing and can be improved by incorporating more accurate aerosol information in
the future. It was noted that the cold biases over mountain areas associated with the
model physics deficiency can not be corrected by aerosol-radiation effects, thus the
correction of aerosol-radiation effect may get complex results and differ with
regions due to the model pre-existing deficiencies.

To quantitatively evaluate the agreement of simulated 2-m temperature with

observations, the mean bias and root mean square error (RMSE) were employed,
and their diurnal variations during the polluted episode averaged over NCP, denoted
by the purple box in Fig. 1a, were displayed in Fig. 9. As shown in Fig. 9a, the warm
bias in NoAero sustained during the entire 24-hr forecast, ranging from 0.3 °C to
0.9 °C. Compared to NoAero, the NCP area-averaged warm bias was remarkably



reduced by ~0.40°C (~73.9%) due to aerosol-radiation interaction, with the
maximum reaching ~0.54 °C (~95.0 %) at 1100 LT (Figs. 9a and c). Consistently
with mean bias, the RMSE was also lower in Aero than NoAero, particularly during
1100 to 2000 LT during the daytime (Figs. 9b and d).

The aerosol-radiation interaction may also have profound impacts on atmospheric

structure in addition to radiation and temperature (Rémy et al., 2015). PBLH is one
of the key parameters to describe the structure of PBL and closely related to air
pollution. It was indicated that the mean daytime PBLH over northern China were
around 300–600m (Fig. 10a), and declined generally 40–200m (10%–40%) in Aero
over the region with highest $PM_{2.5}$ concentration, particularly over Beijing, Tianjin
and Hebei (Figs. 10b–c). As shown in dashed lines in Fig. 11, the NCP
area-averaged PBLH at noon-time (1400 LT) was diminished dramatically by
aerosol-radiation interaction during the pollution event over northern China, with the
maximum decrease reaching -155.2m on 7[th] Dec. The reduction of PBLH could be
the consequence of more stable atmosphere in Aero than NoAero, which was
induced by the terrestrial cooling in the lower part of the planetary boundary layer
and the solar heat due to the absorbing in the upper layers (solid lines in Fig. 11).

The near surface wind fields changes due to aerosol-radiation interaction were

further investigated. Figure 12 shows the wind vector in NoAero (upper panel), Aero
(middle panel) and their difference (lower panel). It can be seen from Fig. 12a-e that
the northern China was dominated by the anticyclonic circulation, accompanied by





the relatively weaker northeast wind over Beijing and Hebei areas. The comparisons
of Aero and NoAero (Figs. 12 k-o) shown that the northeast wind was increased
with the maximum reaching 1 m s$^{-1}$ by aerosol-radiation interaction over Beijing
and Hebei, where high particles concentration located (shadings in Figs. 12 f-j).
Figures 12k-o also indicated the changes of west wind over the south part of the
domain and southeast wind over the ocean areas, which tended to weaken the
anticyclonic circulation and thus declined the wind speed there. The reduced wind
speed due the inclusion of aerosol-radiation interaction was possible due to the
thermal-wind adjustment in response to the more stable near-surface atmosphere,
which was also addressed in previous work using WRF-Chem (Zhang et al., 2015).

The comparisons between simulated wind speeds against in-situ observation

averaged during 6$^{th}$ to 10$^{th}$ Dec. were displayed in Fig. 13. In regard of NoAero, the
simulated wind speed at 10m was overestimated over the nearly whole domain with
the maximum bias up to 3 m s$^{-1}$ except some mountain sites (upper and middle
panels in Fig.13). It might be due to the omission of UCM model as the
overestimation is more prominent in city clusters (especially in Beijing and southern
Hebei) than other areas. Figures 13k-o showed the difference of absolute value of
bias between Aero and NoAero and indicated the bias of simulated wind speed were
decreased over south and northeast part of the domain during afternoon (Figs. 13k-m)
by aerosol-radiation interaction, while were increased over Beijing and Hebei area
particularly during nightfall (Fig. 13n) due to the intensified wind speed there. The



NCP area-averaged bias and RMSE of wind speed at 10m were further shown in
Figure 14. It was seen that the aerosol-radiation interaction helped to reduce the
overestimation of wind speed at 10m up to 0.08 m s$^{-1}$ (~7.8%), particular during
daytime (Figs. 14a and c). Correspondingly, the RMSE of Aero was also lower than
that of NoAero, indicating that the inclusion of aerosol-radiation interaction helped
to improve the prediction of near surface wind speed on the domain-averaged scale.
Although the changes of wind speed is less straightforward than that of radiation,
the aerosol-radiation interactions can also affect dynamic fields (vertical wind shear)
through the changes of atmospheric thermal structure and the thermal wind relation
when the interaction lasts long enough (Huang et al., 2019). Figure 15 displayed
vertical profiles of wind speed at the stations of Beijing and Xingtai in simulation
and verified with sounding observations. It was shown that the NoAero
underestimated (overestimated) the low levels wind speed at 0800 LT (2000 LT) at
both Beijing and Xingtai. However, the wind speed were increased (decreased) at
0800 LT (2000 LT) in Aero relative to NoAero, indicating the positive impacts on
the simulation of atmospheric winds by aerosol-radiation interaction.
Since the forecast meteorological fields by WRF (RMPAS-ST) is routinely
applied to WRF-Chem (RMAPS-Chem) as meteorological ICs in the air quality
operational system at IUM, the changed meteorology due to aerosol-radiation
interaction will further influence the forecast of pollution through meteorological
ICs. In regard of further feedback of aerosol-radiation interactions to the transport



and dissipation of the pollutants, their impacts on wind field at 850hPa were further
discussed as it is strongly correlated with haze formation (Zhang et al., 2018; Zhai et
al., 2019). Figures 16 a-e display that northern China was dominated by the
anticyclone circulation at 850hPa, associated with the southwest (northwest) wind in
the west (east) of the northern part of the domain. The difference of U (zonal,
eastward is positive) winds between Aero and NoAero (middle panel in Fig. 16)
showed that the U wind was intensified over west Hebei, accompanied by the quite
small changes in Beijing area, indicating that the increased U wind was blocked by
the mountains and could not transport the pollutants over Hebei and Beijing to the
east (Figs. 16 f-h). On the other hand, the changes of V (meridional, northward is
positive) show different patterns over north and south of the 38° N (lower panel in
Fig. 16). In the south part, the increased northward wind due to aerosol-radiation
interaction may help to transport pollutants from highly polluted areas to Hebei and
Beijing. In the north of the domain, the negative (positive) changes of V wind
indicated the reduced northward (southward) wind in west (east) of Hebei, which
could suppress the diffusion of the pollutants. As a result, both U and V changes
induced by the aerosol-radiation interaction will prevent pollutants from dispersing
and may exacerbate the pollution in Heibei and Beijing, which confirms the more
stable boundary layer due to aerosol-radiation interaction as discussed earlier.
**4.  Concluding remarks**
To facilitate the future inclusion of aerosol-radiation interactions in the regional


operational NWP system – RMAPS-ST (adapted from WRF) at IUM, CMA, the
impacts of aerosol-radiation interactions on the forecast of surface radiation and
meteorological parameters during a heavy pollution event (Dec. 6th -10th, 2015) over
northern China were investigated. The aerosol information (550-nm AOD 2D field)
were simulated by WRF-Chem and then offline-coupled into RRTMG radiation
scheme of WRF to enable the aerosol-radiation feedback in the forecast. Two sets of
24-hour forecasts were performed at 00UTC from Dec. 2nd-11th, 2015. The only
difference between the two sets of forecasts was whether the aerosol radiative
feedback was activated (Aero) or not (NoAero), while the other schemes remained
the same.
The capability of WRF-chem to reproduce the polluted episode was confirmed
first before the offline-coupling of AOD to WRF. The results indicated that the
temporal variations of simulated AOD at 550nm was in consistent with AERONET
and in-situ observation at IAP. Furthermore, the spatial distributions of $PM_{2.5}$ as well
as their magnitude, particularly during the peak stage (8th to 9th) of the pollution
event were reasonably captured by WRF-Chem, with the correlation coefficients of
0.85, 0.89 and 0.76 at Beijing, Shijiazhuang and Tianjin, respectively.
Further, the impacts of aerosols-radiation interaction on the forecasts of surface
radiation, energy budget, and meteorology parameters were evaluated against
surface and sounding observations. The results showed that the decrease of
downward SW radiation reaching surface induced by aerosol effects helped to



reduce the overestimation of SW radiation during daytime. Moreover, the simulated
surface radiation budget has also been improved, with the biases of net radiation at
surface decreased by 85.3%, 50.0%, 35.4%, and 44.1% during daytime at Beijing,
Tianjin, Taiyuan and Jinan respectively, accompanied by the reduction of sensible
(16.1 W m$^{-2}$, 18.5%) and latent (6.8 W m$^{-2}$, 13.4%) heat fluxes emitted by the
surface at noon-time.

The energy budget changed by aerosol extinction further cools 2-m temperature

by ~0.40°C over NCP, reducing warm bias by ~73.9% and also leading to lower
RMSE, particularly during daytime. Since aerosol cools the lower planetary
boundary layer and meanwhile warms the high atmosphere, it induced the more
stable stratification of the atmosphere and the declination of PBLH by 40–200m
(10%–40%) over NCP. Associating with the changes of planetary boundary structure
and more stable near-surface atmosphere, the aerosol-radiation interaction tended to
weaken the anticyclonic circulation including the east wind over the south part of
the domain and northwest wind over the ocean areas. Thus the bias of wind speed
over south and northeast part of the domain were decreased particularly during the
afternoon, while increased over Beijing and Hebei area. In regard of NCP-average,
the overestimated 10m wind speed was improved during whole day with the
maximum up to 0.08 m s$^{-1}$ (~7.8%) at 1400LT. The comparison between simulated
vertical profiles of atmospheric wind speed with soundings also indicated that Aero
was in better agreement with observation and aerosol-radiation interaction helped to



improve the prediction of dynamic fields such as atmospheric wind through the
thermal wind relation by altering the atmospheric structure.

The impacts of aerosol-radiation interactions on wind field at 850hPa were

further discussed. The results showed that aerosol-radiation interaction will prevent
pollutants from dispersing and may exacerbate the pollution through changes of both
U and V wind, which confirms the more stable boundary layer due to
aerosol-radiation. These wind field changes may also influence the forecast of the
transport and dissipation of the pollutants by WRF-Chem through changed
meteorological ICs.

This study analyzed the impacts of aerosol-radiation interaction on radiation and

meteorological forecast by using the offline-coupling of WRF and high-frequent
updated AOD simulated by WRF-Chem, which is more computationally economic
than the online simulation with the integration time for 96h forecast of about 40% of
that for online simulation. This approach allows for a potential application to include
aerosol-radiation interaction in our current operational NWP systems. The results
revealed that aerosol-radiation interaction had profound influence on the
improvement of predictive accuracy and the potential prospects for its application in
regional NWP in northern China. Given that most of these analyses were based on a
single case of pollution occurred during the wintertime of 2015, there is clearly a
need for further research on more polluted cases to achieve more quantitative results
before the operational application. As the simulated AOD was adopted in the present



study, it should be noted that there exits a discrepancy between simulated AOD and
observation in both spatial distribution and temporal variation, which may influence
the impacts of aerosol-radiation interaction. Meanwhile, surface energy budget and
atmospheric dynamics are also influenced by aerosol-cloud interaction, which are
related to cloud microphysical processes and are not discussed in this study.

***Author contribution*** Yang Yang, Xiujuan Zhao and Dan Chen designed the
experiments and Yang Yang performed the simulations and carried them out. Yang
Yang prepared the manuscript with contributions from all co-authors.

***Acknowledgments*** This work was jointly supported by the National Key R&D
Program of China (grant nos. 2017YFC1501406 and 2018YFF0300102), Natural
Science Foundation of Beijing Municipality (8161004), the National Natural Science
Foundation of China (grant nos. 41705076, 41705087 and 41705135) and
Beijing Major Science and Technology Project (Z181100005418014).



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





Table 1. The variables, sources, numbers of sites in the domain/NCP and the
frequency of chemical and meteorological observations.

| Variables | Source of observation | Numbers of sites over the domain/NCP | Frequency | locations |
|---|---|---|---|---|
| AOD | AERONET | 3/3 | hourly | black dots in Fig. 1b |
| AOD | IAP station | 1/1 | hourly | blue dot in Fig. 1b |
| $PM_{2.5}$ | China National Environmental Monitoring Centre | 813/332 | hourly | dots in Fig. 3a |
| radiation | China Meteorological Administration | 4/4 | hourly | triangles in Fig. 1a |
| radiation | IAP station | 1/1 | hourly | triangles in Fig. 1a |
| 2-m temperature | China Meteorological Administration | 1157/534 | hourly | dots in Fig. 8a |
| wind at 10m | China Meteorological Administration | 1157/534 | hourly | dots in Fig. 8a |
| atmospheric wind | China Meteorological Administration | 2/2 | 0800LT, 2000LT | circles in Fig. 1a |






Table 2. Mean bias of downward SW radiation at surface (W m$^{-2}$) and Net radiation
at surface (W m$^{-2}$) from NoAero and Aero relative to observation during daytime
(averaged 0800 to 1800 LT) and nighttime (averaged 1900 to 0700 LT), averaged
from 6[th] to 11[th] Dec. 2015 at Beijing, Tianjin, Taiyuan and Jinan respectively.

| Station | SW radiation | | Net radiation | | | |
|---------|--------|------|--------|------|--------|------|
| | Daytime | | Daytime | | Nighttime | |
| | NoAero | Aero | NoAero | Aero | NoAero | Aero |
| Beijing | 149.4 | 38.0 | 102.2 | 15.0 | -33.6 | -33.2 |
| Tianjin | 115.5 | 70.9 | 72.2 | 36.4 | -27.1 | -26.4 |
| Taiyuan | 155.0 | 118.3 | 66.9 | 43.2 | -33.6 | -33.3 |
| Jinan | 149.1 | 97.7 | 81.2 | 45.3 | -30.3 | -29.3 |

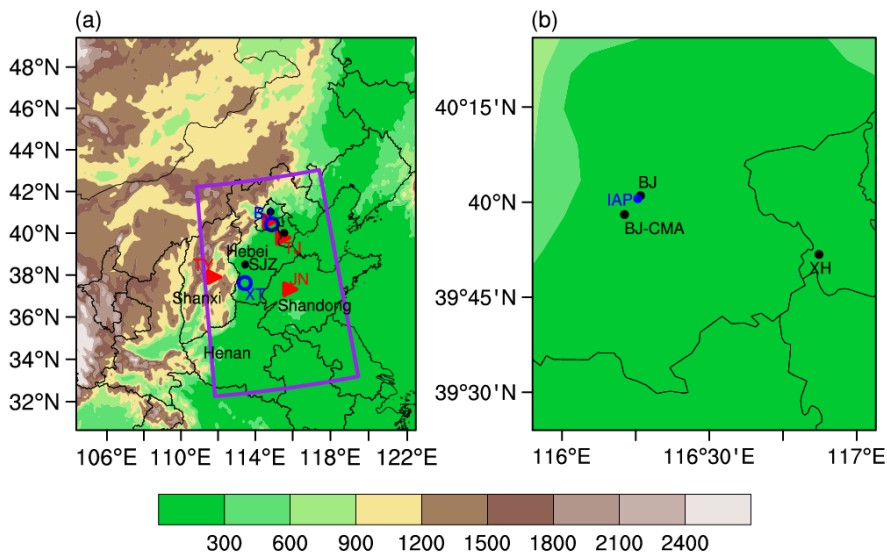

Figure 1. (a) The model domain and the terrain height (shadings, m). Purple box

denotes the NCP, triangles are the observational sites of radiation (BJ: Beijing, TJ:

Tianjin, TY: Taiyuan and JN: Jinan), circles are sites of sounding observation (BJ:

Beijing and XT: Xingtai), dots denote the major cities for validation of $PM_{2.5}$ (BJ:

Beijing, SJZ: Shijiazhuang and TJ: Tianjin). Names of provinces are also added

(Hebei, Shanxi, Shandong and Henan). (b) The observational sites of AOD,

including AERONET sites (black dots, BJ: Beijing, BJ-CMA: Beijing-CMA and XH:

Xianghe) and IAP in-situ (blue dot) site.



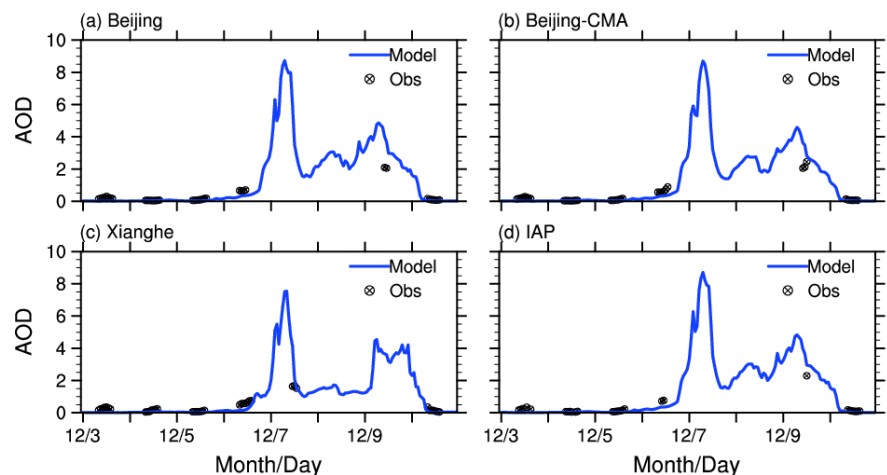


Figure 2. Temporal variation of observed (black dots) and simulated (blue) AOD at

550nm during 3rd-10th Dec. (LT) at (a) Beijing, (b) Beijing-CMA, (c) Xianghe and (d)

IAP, AOD observations are from (a-c) AERONET and (d) IAP in-situ site.



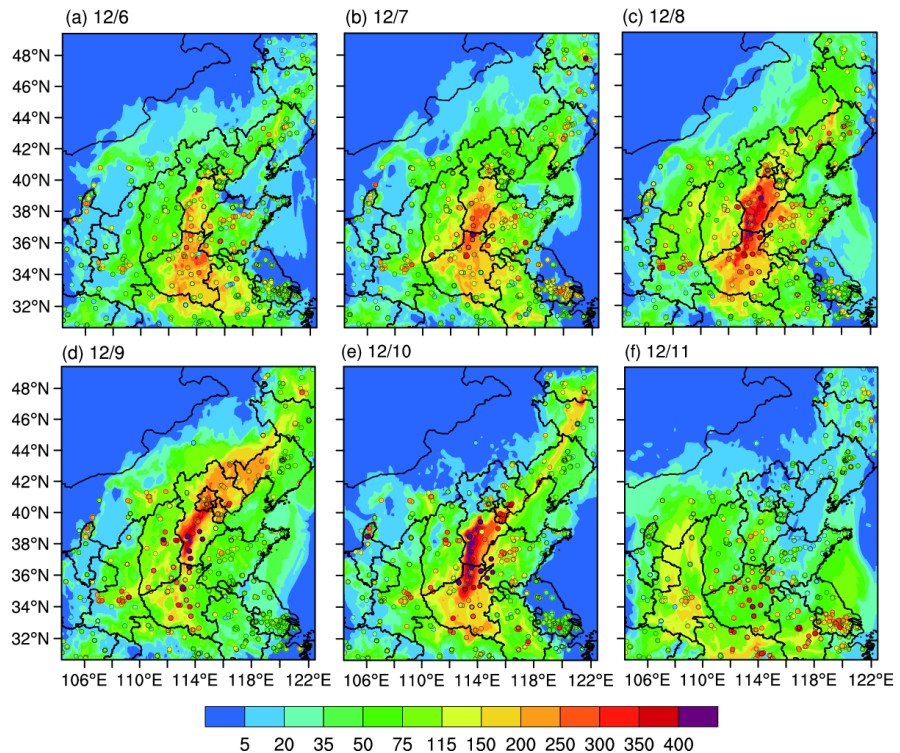


Figure 3. Observed (colored dots) and WRF-Chem simulated (shadings) spatial
distribution of $PM_{2.5}$ concentrations ($\mu g\ m^{-3}$) on 0800LT of (a) 6[th], (b) 7[th], (c) 8[th], (d)
9[th], (e) 10[th] and (f) 11[th] Dec. respectively.



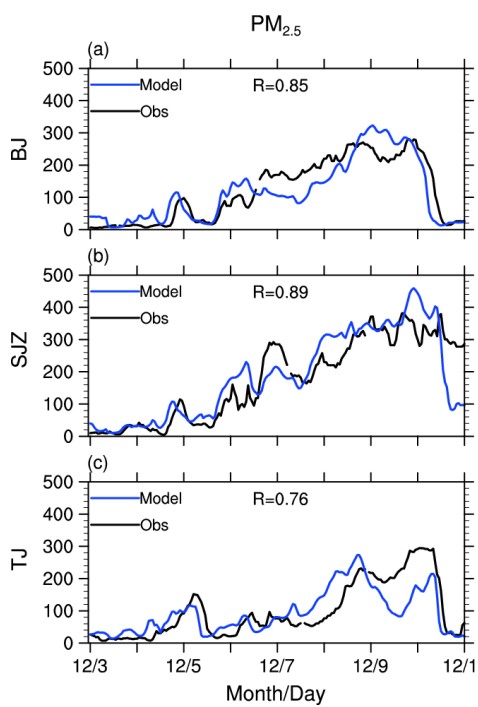


Figure 4.   Observed (black) and WRF-Chem simulated (blue) temporal variation of
$PM_{2.5}$ ($\mu$g m$^{-3}$) at three major cities: (a) Beijing (BJ), (b) Shijiazhuang (SJZ) and (c)
Tianjin (TJ).



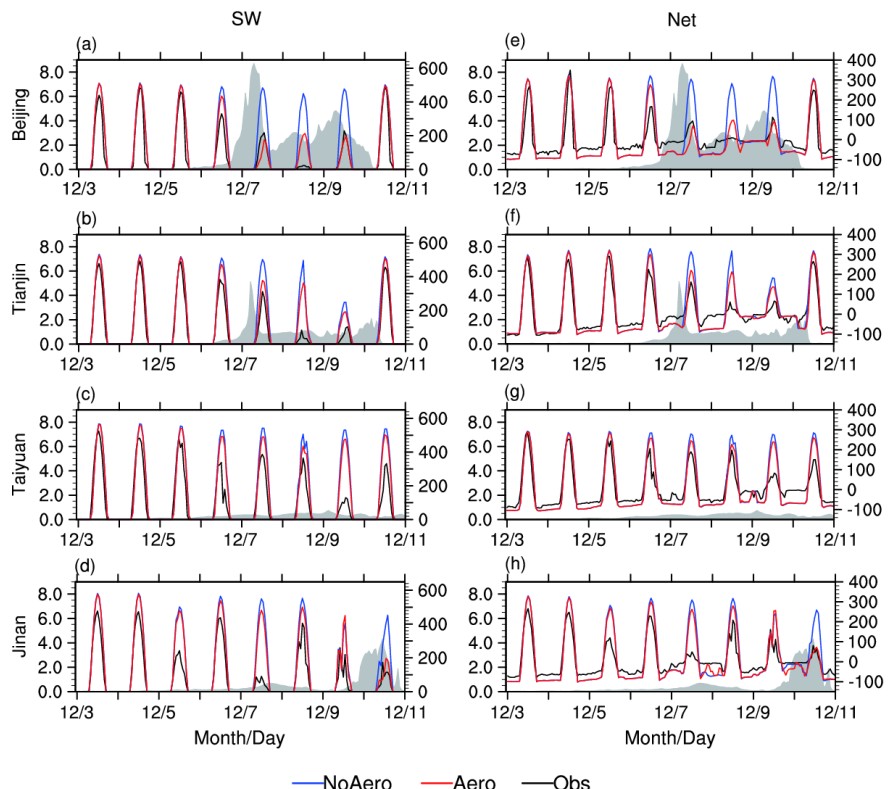


Figure 5. (a–d) observed (black) and WRF simulated (NoAero: blue, Aero: red)


temporal variation of    downward shortwave radaition at surface (W m$^{-2}$, right axis)


at (a) Beijing, (b) Tianjin, (c) Taiyuan and (d) Jinan, respectively. The grey areas


indicate the simulated AOD (left axis) by WRF-Chem. (e–h) are same with (a–d),


but for net radaition at surface (W m$^{-2}$).




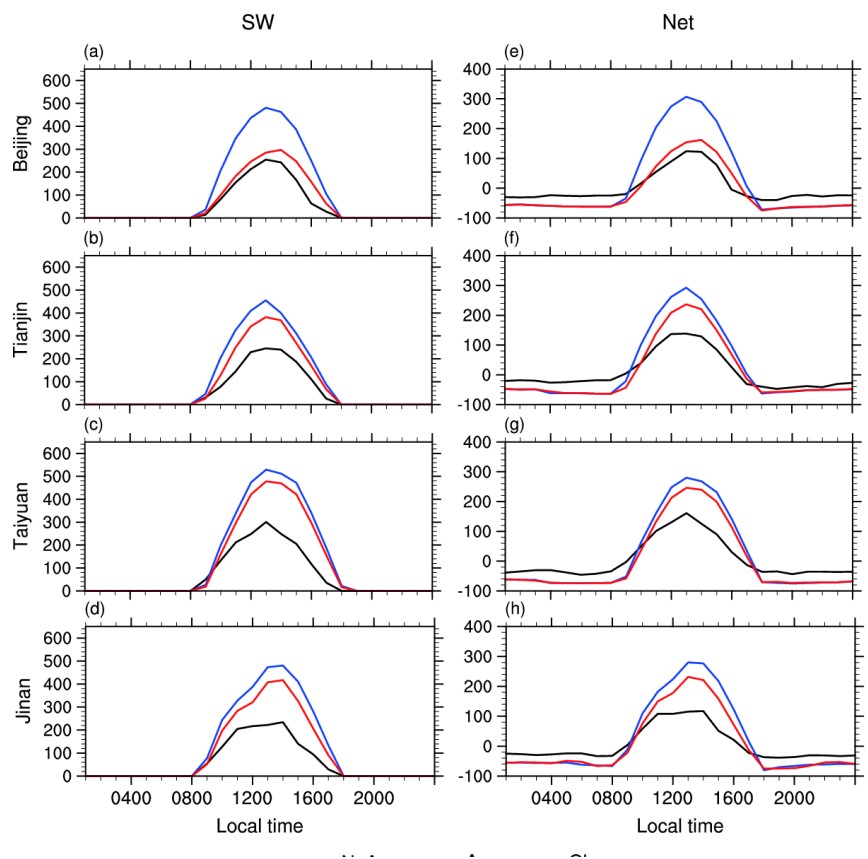


Figure 6. (a–d) observed (black) and simulated (NoAero: blue, Aero: red) diurnal
cycles of downward shortwave radaition at surface (W m$^{-2}$) averaged from 6[th] to 10[th]
Dec. 2015 at (a) Beijing, (b) Tianjin, (c) Taiyuan and (d) Jinan, respectively. (e–h)
are same with (a–d), but for net radaition at surface (W m$^{-2}$).

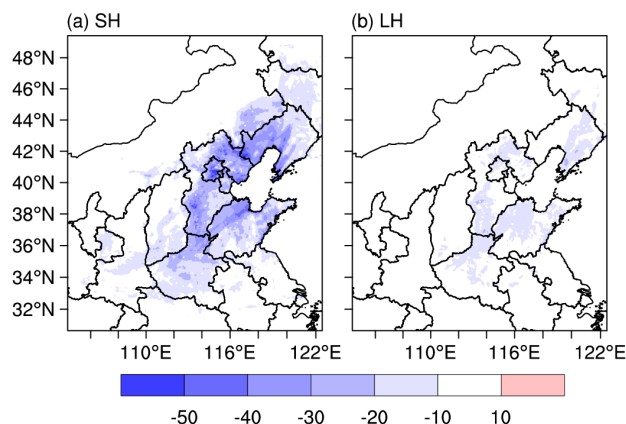


Figure 7. The differences (Aero minus NoAero) of (a) surface sensible heat flux and
(b) surface latent heat flux (W m−2, upward is positive) at 1300LT averaged from
6th to 10th Dec. 2015.



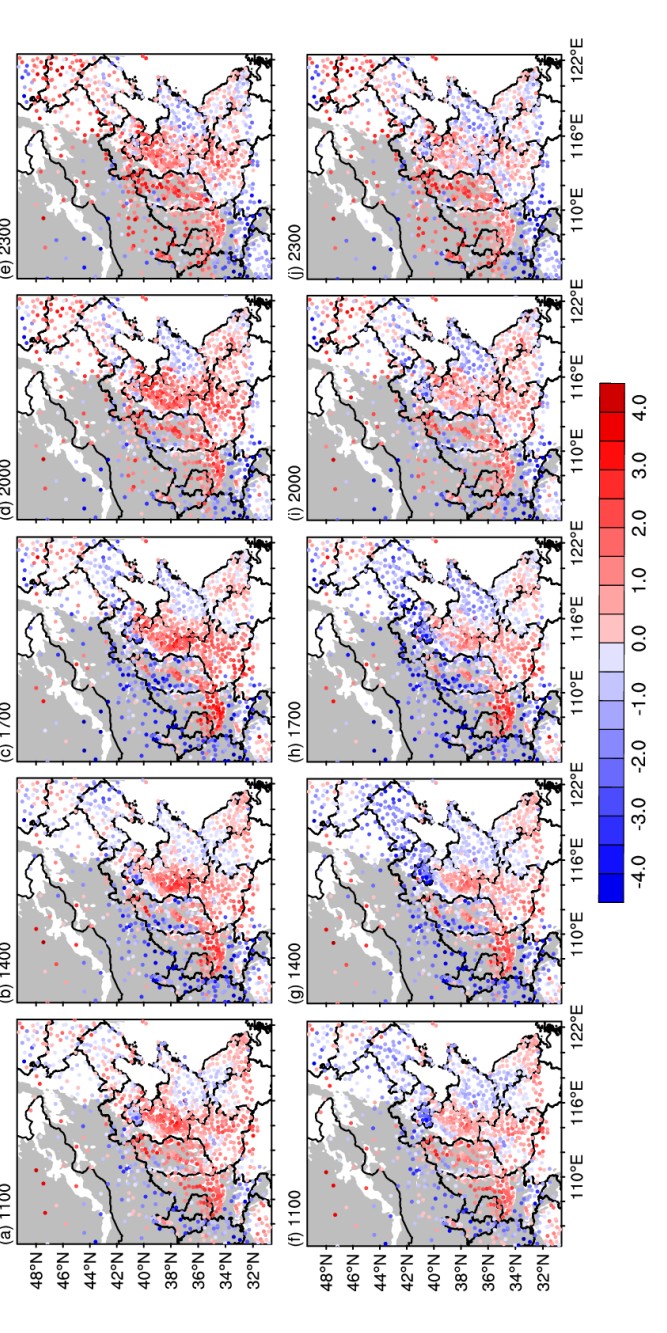

Figure 8. The bias of 2-m temperature (°C) at (a) 1100, (b) 1400, (c) 1700, (d) 2000 and (e) 2300 LT in NoAero averaged from 6th to 10th Dec.

2015, (f–j) are same with (a–e), but for Aero. The grey areas denote the areas of terrain height above 1000m.




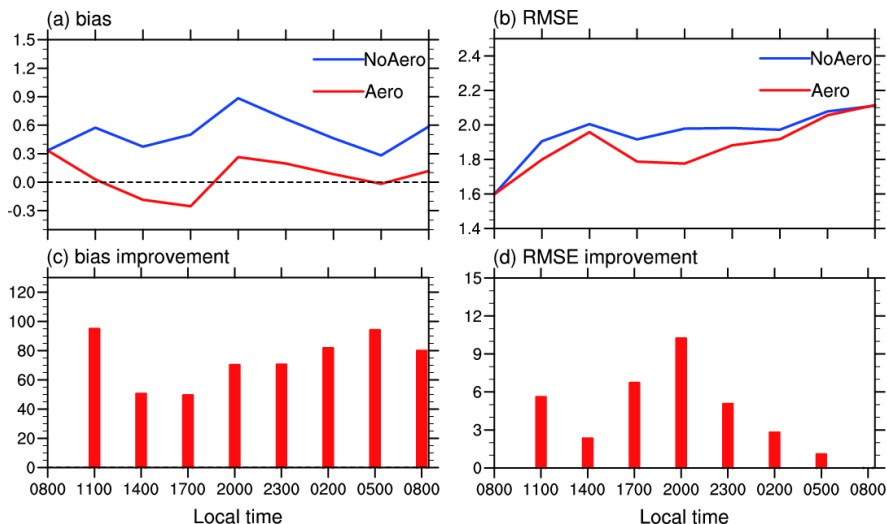

Figure 9. Area-averaged (a) bias and (b) RMSE of simulated 2-m temperature (°C)

in NoAero (blue) and Aero (red) over NCP area (defined in Fig. 1a), averaged from

6th to 10th Dec. 2015, and the mean improvement (%) of (c) absolute value of bias

and (d) RMSE in Aero relative to NoAero.



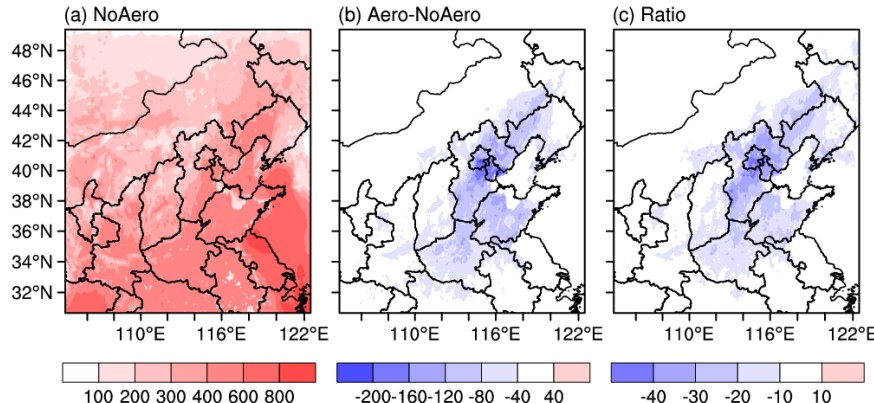


Figure 10. Daytime mean PBLH (m) in NoAero, (b) the difference between Aero
and NoAero (Aero minus NoAero) and (c) the ratio of changes (%) averaged during
6th to 10th Dec. 2015.

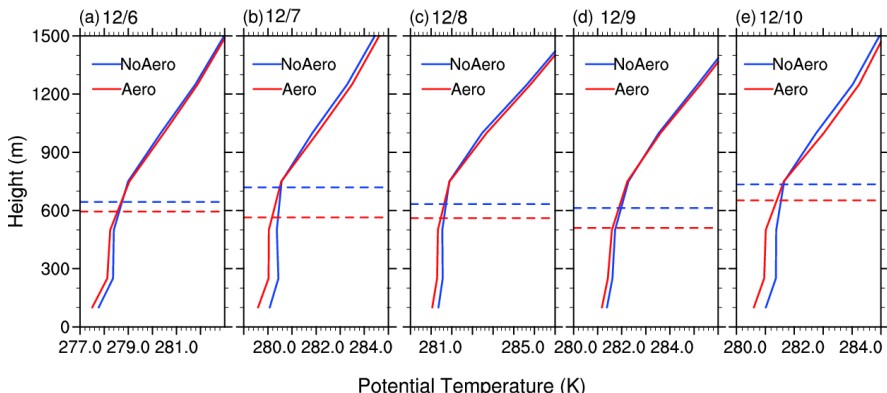


Figure 11. NCP (defined in Fig. 1a) area-averaged vertical profiles of potential
temperature (K, solid) and planetary boundary-layer height (m, dash) in NoAero
(blue) and Aero (red) at 1400 LT of (a) 6th, (b) 7th, (c) 8th, (d) 9th and (e) 10th Dec.

820     2015.

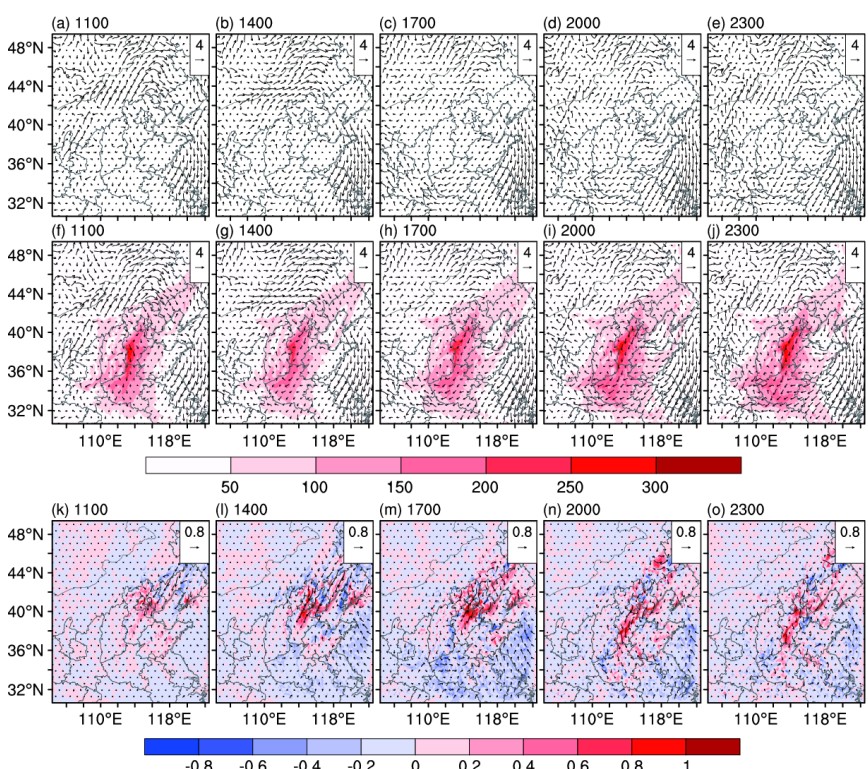

Figure 12. The 10m wind (vector) at 1100, 1400, 1700, 2000 and 2300 LT in (a–e)

NoAero and (f–j) Aero averaged during 6th to 10th Dec. 2015, shadings in (f–j) are

simulated PM$_{2.5}$ concentrations (μg m$^{-3}$). (k–o) the difference of 10m wind (vector)

and wind speed (shadings) between Aero and NoAero (Aero minus NoAero).

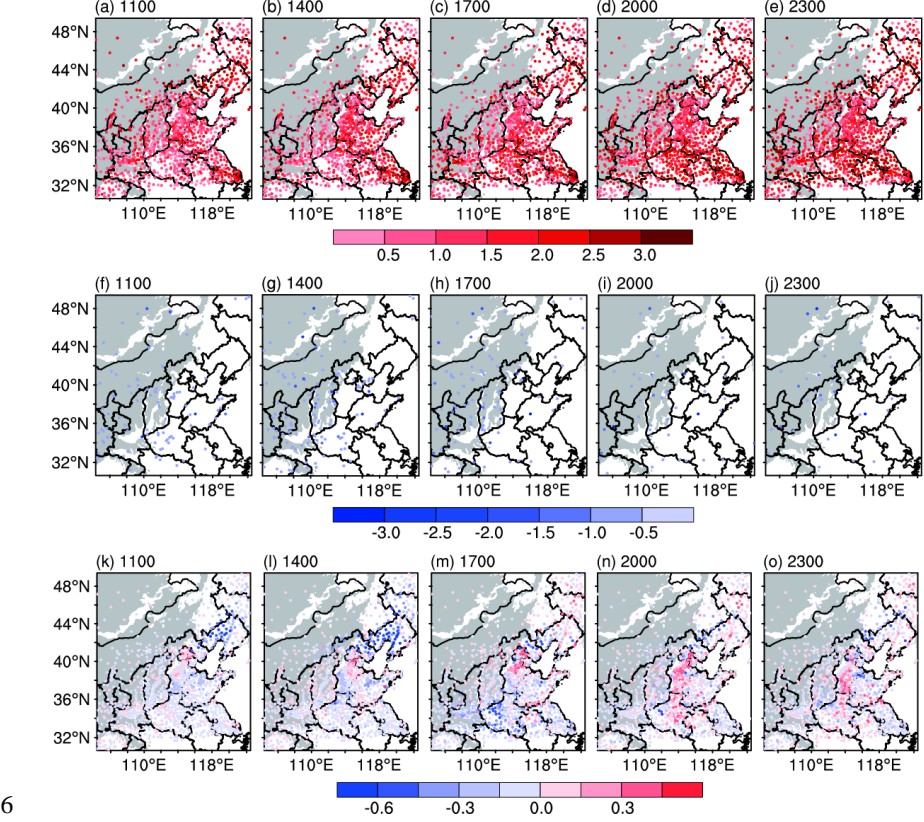

826

Figure 13. The bias of 10m wind speed (m s$^{-1}$) at 1100, 1400, 1700, 2000 and 2300

LT for (a–e) overestimated sites and (f–j) underestimated sites in NoAero averaged

during 6$^{th}$ to 10$^{th}$ Dec. 2015.  (k–o) the difference of absolute value of bias (m s$^{-1}$)

between Aero and NoAero (Aero minus NoAero). The grey areas denote the areas of

terrain height above 1000m.





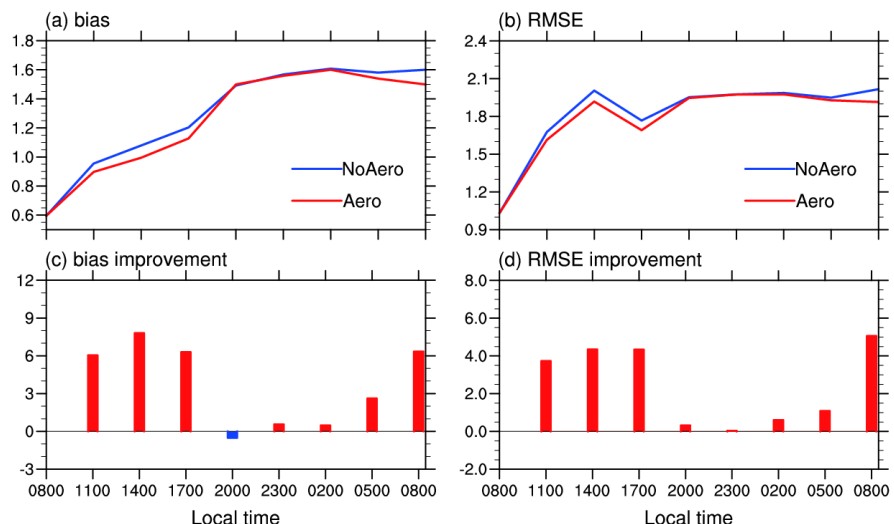

Figure 14. Same with Fig.9, but for wind speed at 10m (m s$^{-1}$).




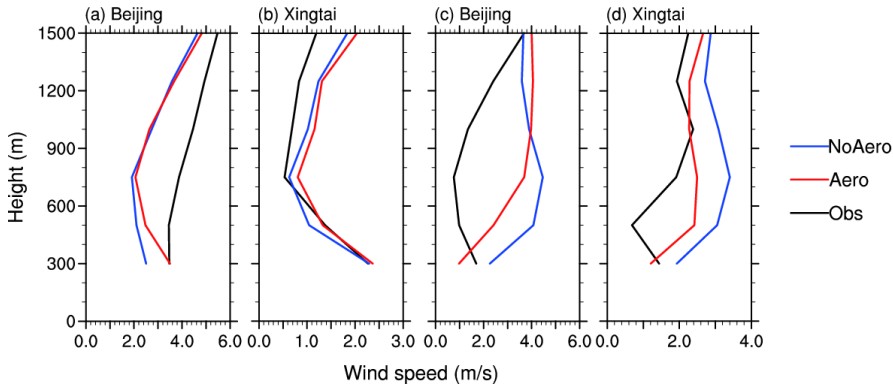

Figure 15. (a–b) Observed (black) and simulated (NoAero: blue, Aero: red) vertical
profiles of atmospheric wind speed (m s$^{-1}$) at (a) Bejing and (b)Xingtai at 0800LT
averaged from 6[th] to 10[th] Dec., (c–d) are same with (a–b), but at 2000LT.

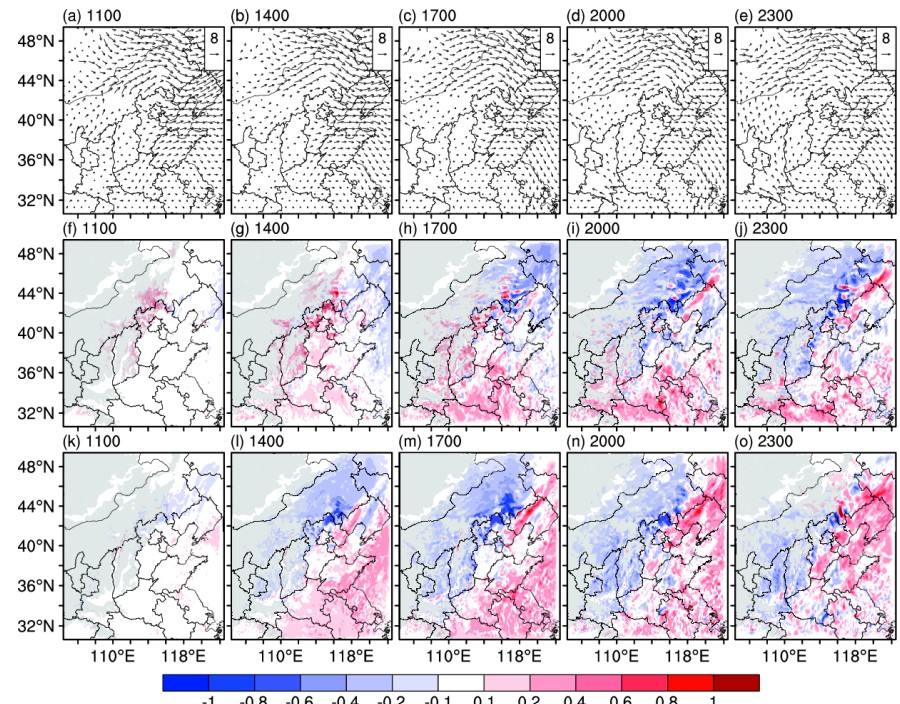


Figure 16. The wind at 850hPa (vector) at 1100, 1400, 1700, 2000 and 2300 LT in

NoAero averaged during 6[th] to 10[th] Dec. 2015. The difference of (f–j) U and (k–o) V

wind speed between Aero and NoAero (Aero minus NoAero). The grey areas denote

the areas of terrain height above 1000m.