# Peer review of "Impacts of aerosol-radiation interaction on meteorological forecast over northern China by offline coupling the WRF-Chem simulated AOD into WRF: a case study during a heavy pollution event"

_Atmospheric Chemistry and Physics, 2019_

## Referee Comment (RC1) · Anonymous Referee #2 · 17 Feb 2020

Review of "Impact of aerosol-radiation interaction on meteorological forecast over northern China by offline coupling the WRF-Chem simulated AOD into WRF: a case study during a heavy pollution event" by Yang et al.

**General summary**

This paper assesses the impact of incorporating aerosol-radiation interactions in the NWP models on surface radiation and weather forecasts during a heavy pollution episode in North China Plain. Hourly AOD fields simulated using WRF-Chem model are fed offline into the radiation schemes of a WRF based NWP system called RMAPS-RT. The inclusion of aerosols in the NWP system reduced overestimation of daytime surface radiation magnitude and budget, and improved forecasts of temperature and wind speed. The results highlight the importance of including aerosols in the NWP system and are interesting. However, the paper lacks detailed evaluation of AOD and PM2.5 (see my specific comments on improving the evaluation part). Additionally, the paper does not discuss whether or not aerosol induced changes in the weather forecast are statistcally significant or not. If changes are not statically signficant, it may not be worthwhile to incorporate more realistic aerosol information in the NWP models and just a climatological aerosol representation in the raidation routines may be sufficient. Thus, I recommend major revisions of the paper before publication in ACP.

**Specific comments**

Line 123: change "accessed" to "assessed".

Line 195-196: why RRTMG was not used for WRF-Chem simulations. Are aerosol-radiation interactions turned off purposely in the WRF-Chem simulations?

Line 203: Why FNL data were used in WRF-Chem experiments and ECMWF data used as met IC/BC in WRF forecast? What is the sensitivity of meteorological parameters to different driving datasets?

Lines 205-206: Did you run WRF-Chem continuously for 10 days? If yes, did you use any kind of nudging to limit the drift of meteorological fields from the large-scale reanalysia fields?

Lines 213-214: I do not agree that MODIS AOD retrievals are not available during this episode. I did a quick average AOD plot in Giovanni and the resulting images are shown below in Figures R1 and R2 for both MODIS Terra and Aqua satellites. While AOD is not available everywhere in the domain but I think the datasets is still useful for validation of the model simulated spatial distribution of AOD. I encourage the authors to use Level 2 MODIS AOD retrievals for comparison with WRF-Chem.

Lines 249-250: In addition to my above comment, the authors should consider using other satellite-based products such as MISR and MAIAC AOD, and aerosol extinction coefficient retrievals from CALIPSO.

[Figure]

**Figure R1:** Time averaged MODIS Terra AOD map for 6-9 December 2015.

[Figure]

**Figure R2:** Time averaged MODIS Aqua AOD map for 6-9 December 2015.

Figure 3 and Lines 253-259: This discussion is very qualitiative and I recommend the authors to include some quantitative information about the evalaution. I suggest plotting time series of

hourly averaged observed and modeled PM2.5 mass concentrations over the Henen and Hebei provinces (similar to Fig. 4 for the three cities). Maps of bias, root mean square error, and correlation coefficient for each site for the heavy pollution and cleaner periods will also be useful to understand model skill in reproducing the heavy pollution event.

Line 279: Change "were overlay" to "were overlaid".

Figure 5: Why does the AOD peak before the reduction in SW especially on 6th June? At Taiyuan, there is not much difference between Aero and NoAero simulations which may be because AOD at this site is likely not captured well by the model.

Line 351: change "biases" to "biased".

Line 355: change "leaded" to "led"

Line 391: change "shown" to "showed".

Section 3.2.2 and related figures: Are the changes in different meteorological parameters statistically significant?

---

## Referee Comment (RC2) · Angela Benedetti (Referee) · 23 Mar 2020

**General comments**

The article is interesting and treats a topic of utmost relevance, that of aerosol impacts on Numerical Weather Prediction (NWP). The authors have analysed in great detail a pollution case in Northern China during December 2-11, 2015 and examined the impact of including aerosol radiative forcing on several key meteorological vari-

ables. They found that aerosols have a large impact on shortward radiative fluxes at the surface and consequently on 2m temperatures and wind speed using independent observations from various networks to establish that. These results are consistent with finding from other authors who highlighted the importance of a correct inclusion of aerosol fields particularly under extreme aerosol loads.

The paper deserves attention and with some refinements will be acceptable for publication. However, it is worthwhile to stress that case studies such as this may not be statistically significant, especially because extreme aerosol conditions were chosen. It would be necessary to run more cases, possibly entire seasons. I would encourage the authors to get in touch with the rest of the community and join an effort sponsored by WMO via various committees (WGNE, GAW and S2S) to run coordinate experimentation in regional and global models with the goal to gain a fuller picture of the aerosol impacts in NWP. Feel free to contact me directly about this.

**Minor comments and typos**

line 22, high-frequency

line 66, episodic aerosol events

ine 105, to facilitate the inclusion of...

line 116, was included

line 119 For these research studies using operational NWP systems, offline approaches were mostly used. Actually, in Remy et al 2015 and Mulcahy et al 2014 that was not the case and the interactive aerosols were run online.

line 143, in an NWP system

line 152, future applications

line 153, The remainder of the paper is organised... Please change all tenses in this paragraph to present.

line 168, National

line 169, Environmental - please re-run the paper through a spell and grammar checker to ensure that typos are corrected

line 171, with a higher

line 174 the Rapid Radiative

line 181 The RRTMG

line 185 was input

line 186 integral

lie 189 which was - please check that verbs are correctly conjugated

line 190 the same configuration

line 206 did you investigate the sensitivity of the model AOD to the choice of these ICs and BCs?

line 216 were CARSNET (https://www.atmos-chem-phys.net/15/7619/2015/) observations available over the area? if yes, why were they not used?

line 237 / Figure 2 I think it would be good to have extra data from CARSNET if possible

line 244 most of them, do you mean the observations during the peak? See comment above.

line 245 were similar to

line 247 you need more observations to establish that

line 265 do you think this was because of the emission inventories used or the skill of the model or both? Please comment.

line 286 In the NoAero experiments were the aerosols completely missing from the

simulation or was a climatology used?

line 302 if a climatology were used would this discrepancy be less severe? I am assuming that in the NoAero simulations there were really no aerosols.

line 304 this type of bias in SW fluxes is huge

line 321/Figure 6 At some stations the bias in SW fluxes is not improved as much as in Beijing - do you have an explanation for that?

line 341 are discussed

line 368 is this an average value? With the biases in SW radiation being so large I would have expected higher temperature biases.

line 420 / Figure 15, the wind profile at Beijing is quite different from observations in both Aero and NoAero experiments, do you have an explanation for that?

line 450 very nice discussion of the impacts on the vertical stratification

line 461, please specify if an aerosol climatology was used in NoAero

line 520 the fact that aerosol-cloud interactions were not included in the study should be mentioned also at the beginning

---

## Author Comment (AC1) · 18 May 2020

We have addressed all the comments raised by the reviewer, and incorporated them in the revised manuscript. Please find attachment our itemized responses to the reviewer's comments.

Please also note the supplement to this comment:

[Figure]

https://www.atmos-chem-phys-discuss.net/acp-2019-1056/acp-2019-1056-AC1-supplement.pdf

---

## Author Response (AR1)

**Dear ACP Editor:**

**We have addressed all the comments raised by both reviewers, and incorporated them in the revised manuscript. Please find below our itemized responses to the reviewer's comments.**

**Thank you very much for your consideration.**

**Sincerely,**
**Yang Yang, et al.**
* * *
**COMMENTS TO THE AUTHOR(S)**

Impacts of aerosol-radiation interaction on meteorological forecast over northern China by offline coupling the WRF-Chem simulated AOD into WRF: a case study during a heavy pollution event
Manuscript ID: acp-2019-1056
Authors: Yang, et al.

**Reviewer 1**
**General summary**
This paper assesses the impact of incorporating aerosol-radiation interactions in the NWP models on surface radiation and weather forecasts during a heavy pollution episode in North China Plain. Hourly AOD fields simulated using WRF-Chem model are fed offline into the radiation schemes of a WRF based NWP system called RMAPS-RT. The inclusion of aerosols in the NWP system reduced overestimation of daytime surface radiation magnitude and budget, and improved forecasts of temperature and wind speed. The results highlight the importance of including aerosols in the NWP system and are interesting. However, the paper lacks detailed evaluation of AOD and PM2.5 (see my specific comments on improving the evaluation part). Additionally, the paper does not discuss whether or not aerosol induced changes in the weather forecast are statistically significant or not. If changes are not statically significant, it may not be worthwhile to incorporate more realistic aerosol information in the NWP models and just a climatological aerosol representation in the radiation routines may be sufficient. Thus, I recommend major revisions of the paper before publication in ACP.

**Response:**

We really appreciate the valuable comments. We have made the following changes according to these comments.

More detailed evaluation of simulated AOD against MODIS and CALIPSO satellite-based products were performed and added in the revised manuscript. In addition, we added more quantitative evaluations of $PM_{2.5}$ mass concentrations including spatial distributions of bias, root mean square error, and correlation coefficient for individual sites during pollution and relatively cleaner periods, as well as the time series of hourly averaged observed and simulated $PM_{2.5}$ concentrations over the Henan and Hebei provinces.

To address the issue about the statistically significance of the aerosol induced impacts on weather forecast, we further conducted three sets of 24-hour forecasts for a longer period lasting 27 days (Jan. 13th- Feb. 8th, 2017), with no AOD field (NoAero), climatological AOD fields (ClimAero) and WRF-Chem simulated hourly AOD fields (ChemAero) included, respectively. The results indicated that the simulation with the inclusion of WRF-Chem simulated hourly AOD fields outperformed other two simulations and showed more improvement on the forecast of surface temperature and near surface wind speed than the simulation with climatological AOD fields. These results are in consistent with the conclusions in the current study. Please see more detailed discussion below.

We expect that you will find that your comments have been considered fully and properly in our revised manuscript. Below are our item-by-item responses.

**Specific comments**

Line 123: change "accessed" to "assessed".
**Response:**

Thanks, corrected.

Line 195-196: why RRTMG was not used for WRF-Chem simulations. Are aerosol-radiation interactions turned off purposely in the WRF-Chem simulations?

**Response:**

Thanks for your insightful comment. The RRTMG scheme was not included in the version 3.3.1 of WRF-Chem, which was applied in the current study and also in our operational system. The aerosol-radiation interactions were turned off in the WRF-Chem simulations. We do understand that the aerosol-radiation interactions could benefit the simulation of $PM_{2.5}$, particularly the peak values. We would include the aerosol-radiation interactions of WRF-Chem in online test in our further research.

Line 203: Why FNL data were used in WRF-Chem experiments and ECMWF data used as met IC/BC in WRF forecast? What is the sensitivity of meteorological parameters to different driving datasets?

**Response:**

Thanks for your comment. The ECMWF forecast data were adopted as meteorological IC/BC in the operational meteorological forecast system based on WRF, and the meteorological field forecasted by WRF with the inclusion of data assimilation were then input as IC/BC of operational WRF-Chem simulation. In the beginning of the current study, we first tried FNL data for meteorological IC/BC of WRF-Chem forecast and found that the results were reasonable and satisfying, so we did not evaluate the sensitivity of meteorological driving datasets further. According to the colleagues in our Development Testbed Center, the direction of the WRF forecast meteorological parameter biases (e.g. overestimated temperature) are not so sensitive to the initial conditions, as the same direction of biases occur quickly even with assimilated initials that intentionally overcorrect the original biases; thus we assumed the system is more sensitive to certain processes instead of initials in current configurations. We will conduct further detailed research and test about the sensitivity of meteorological parameters to different driving datasets in our future research.

Lines 205-206: Did you run WRF-Chem continuously for 10 days? If yes, did you use any kind of nudging to limit the drift of meteorological fields from the large-scale reanalysis fields?

**Response:**

Thanks for your comment. We tried without nudging over the plain areas of northern China during wintertime in our previous study, and found that the simulations of pollutant were reasonable and the drift were acceptable. Thus, we run WRF-Chem continuously for 10 days without nudging in the current study.

Lines 213-214: I do not agree that MODIS AOD retrievals are not available during this episode. I did a quick average AOD plot in Giovanni and the resulting images are shown below in Figures R1 and R2 for both MODIS Terra and Aqua satellites. While AOD is not available everywhere in the domain but I think the datasets is still useful for validation of the model simulated spatial distribution of AOD. I encourage the authors to use Level 2 MODIS AOD retrievals for comparison with WRF-Chem.

[Figure]

Figure R1: Time averaged MODIS Terra AOD map for 6-9 December 2015.

[Figure]

Figure R2: Time averaged MODIS Aqua AOD map for 6-9 December 2015.

**Response:**

Thanks for your helpful and insightful comment. According to your suggestion, we evaluated the simulated AOD with MODIS Terra and Aqua (Fig. S1). It was seen that WRF-Chem is capable to capture the AOD spatial distribution and also reproduced the transport paths during the event. The simulated high-valued AOD located in Henan on Dec. 6[th], then the center moved to Hebei and Beijing on 7[th] and shifted to northeast areas afterwards. The variations of simulated AOD were in consistent with both Terra and Aqua with slightly overestimated peak value of AOD. In particular, the simulated shifting of AOD center to northeast areas was also observed in Aqua (Fig. S1r-s). We have added the Fig. S1 and the discussion in the revised manuscript (around L247-L255).

[Figure]

Fig. S1 The WRF-Chem simulated and MODIS observed spatial distribution of AOD on 6[th]-

10th December (from left to right). The first (a-e) and third rows (k-o) are WRF-Chem simulations at 1000LT and 1300LT (MODIS path times) respectively. The second (f-j) and fourth (p-t) rows are MODIS Terra and Aqua observations, respectively. Gray areas in (f-j) and (p-t) denote the missing values.

Lines 249-250: In addition to my above comment, the authors should consider using other satellite-based products such as MISR and MAIAC AOD, and aerosol extinction coefficient retrievals from CALIPSO.

**Response:**

Thanks for your helpful and insightful comment. According to your suggestion, we have compared the modeled 550nm aerosol extinction coefficient with CALIPSO, and displayed AOD from MISR level 3 daily product.

Fig. S2 displayed the vertical distribution of simulated 550nm aerosol extinction coefficient compared to those from CALIPSO. Four cross sections along CALIPSO paths on 6th to 9th December were shown. The results indicated that the model could generally reproduce the vertical distribution of extinction coefficients at 550nm in terms of comparable magnitude with those from CALIPSO, particularly on 6th, 7th and 9th, December. However, CALIPSO showed more high values at lower altitude (below 1km) that model failed to capture; the inconsistency may be associated with both CALIPSO retrieval uncertainties at the low altitude and the model itself. We have added the Fig. S2 and the discussion in the revised manuscript (around L255 – L264).

Fig.S3 showed the spatial distribution of AOD at 555nm during 6th to 10th December obtained from MISR. It was seen that the valid fields of AOD from MISR are quite limited during this polluted episode. Therefore, we evaluated the simulation of AOD against MODIS AOD (as discussed earlier) rather than MISR.

[Figure]

Fig. S2 The WRF-Chem simulated 550nm AOD (shadings) on (a)1800UTC of 6[th], (b) 0400UTC of 7[th], (c)1700UTC of 8[th], (d) 0400UTC of 9[th] December overlaid with CALIPSO paths (black thick solid). (e-l) denote the corresponding vertical distributions of aerosol extinction coefficient at 550nm from (e, g, i, k) CALIPSO and (f, h, j, l) model simulations. Gray areas in (e-l) denote the terrain.

[Figure]

Fig. S3 The spatial distribution of AOD at 555nm from MISR on (a) 6[th], (b) 7[th], (c) 8[th], (d) 9[th] and (e) 10[th] December respectively, the gray areas indicate the missing values.

Figure 3 and Lines 253-259: This discussion is very qualitative and I recommend the authors to include some quantitative information about the evaluations. I suggest plotting time series of hourly averaged observed and modeled PM2.5 mass concentrations over the Henen and Hebei provinces (similar to Fig. 4 for the three cities). Maps of bias, root mean square error, and correlation coefficient for each site for the heavy pollution and cleaner periods will also be useful to understand model skill in reproducing the heavy pollution event.

**Response:**

Thanks for your helpful suggestions. We have added the spatial distributions of bias, root mean square error, and correlation coefficient for individual site during the heavy pollution and relatively cleaner periods (Fig. S4), and the time series of hourly averaged observed and modeled $PM_{2.5}$ mass concentrations over the Henan and Hebei provinces (Fig. 5 d-e).

Figure S4 displayed the mean bias, root mean square error (RMSE), and correlation coefficient during the heavy pollution and relatively cleaner periods. It was seen that the biases of $PM_{2.5}$ were generally less than 40 μg m$^{-3}$ with the correlation coefficient exceeding 0.8 during clean period (Fig. S4a-c). Compared with clean period, the bias and RMSE were generally larger during polluted period (Fig. S4d-f). The $PM_{2.5}$ concentrations over most areas of the domain were underestimated with the maximum bias exceeding 160 μg m$^{-3}$. Overall, the correlation coefficient was generally higher than 0.4 in northern China during the polluted period, particularly over Beijing with the correlation coefficient reaching 0.8.

To further assess the temporal evolutions of the pollution, the simulated $PM_{2.5}$ concentrations at three major cities (Beijing, Shijiazhuang and Tianjin) and two provinces (Hebei and Henan) in northern China were compared with observation as shown in Fig. S5. It showed that the hourly variations of PM$_{2.5}$ concentration, including the occurrence of several high peaks at the three cities, as well as the gradual accumulation of pollution in Hebei and Henan could be reasonably reproduced by WRF-Chem. The correlation coefficients (R) between simulation and observation at Beijing, Shijiazhuang, Tianjin, Hebei and Henan were 0.85, 0.89, 0.76, 0.92 and 0.77 respectively. It should be noted that there exits slight overestimation (underestimation) of the peak magnitude during 9$^{th}$ to 10$^{th}$ at Beijing and Shijiazhuang (Tianjin, Hebei and Henan); the overestimation in Beijing and Shijiazhuang is possibly associated with the frequent emission changes caused by emission-control-measures in reality which are not dynamically updated in the model; the underestimation is more related with the deficiency of model skills, such as missing heterogeneous reaction paths in the chemistry scheme.

We have added Fig. S4-5 and the corresponding statement in the revised manuscript around L284-L293 and L294-L308.

[Figure]

Fig. S4. The (a, d) bias (μg m$^{-3}$), (b, e) RMSE (μg m$^{-3}$), and (c, f) correlation coefficient (1) averaged (a-c) during clean period (3$^{th}$ to 5$^{th}$ Dec) and (d-f) the polluted period (6$^{th}$ to 10$^{th}$ Dec).

[Figure]

Fig. S5. Observed (black) and WRF-Chem simulated (blue) temporal variation of PM$_{2.5}$ (μg m$^{-3}$) at three major cities (a) Beijing (BJ), (b) Shijiazhuang (SJZ) (c)Tianjin (TJ) and two provinces (d) Hebei (HB) and (e) Henan (HN).

Line 279: Change "were overlay" to "were overlaid".

**Response:**

    Thanks, corrected.

Figure 5: Why does the AOD peak before the reduction in SW especially on 6th June? At Taiyuan, there is not much difference between Aero and NoAero simulations which may be because AOD at this site is likely not captured well by the model.

**Response:**

Thanks for your comment. The pollutant started to accumulate since 6th Dec., accompanied by the increment of AOD. However, the impacts of aerosol-radiation interactions on meteorological fields mainly occurred during daytime through its direct influence on radiation, especially shortwave (SW) radiation. Therefore, the peak timing of reduction in SW may not coincide exactly with that of AOD. In addition, the relation between AOD-induced radiation changes (through aerosol-radiation interaction) and AOD value is not linear.

Line 351: change "biases" to "biased".

**Response:**

Thanks, corrected.

Line 355: change "leaded" to "led"

**Response:**

Thanks, corrected.

Line 391: change "shown" to "showed".

**Response:**

Thanks, corrected.

Section 3.2.2 and related figures: Are the changes in different meteorological parameters statistically significant?

**Response:**

Thanks for your comment. To addressed this issue, we conducted three sets of 24-hour forecasts for a longer period lasting 27 days (Jan. 13th – Feb. 8th, 2017), with no AOD field (NoAero), climatological AOD fields (ClimAero) and WRF-Chem simulated hourly AOD fields (ChemAero) included, respectively.

The results indicated that the temperature was underestimated (overestimated)

during daytime (nighttime) in NoAero experiment. The temperature is reduced by the aerosol-radiation interactions by inclusion of either climatological or WRF-Chem simulated AOD fields (Fig. S6a), which tends to increase the bias during daytime, and decrease the bias during nighttime. However, the RMSE of temperature in ChemAero is lower than NoAero during the whole 24-hr forecast, particularly at 2000LT of nightfall with the reduction of RMSE reaching ~ 9%. While the RMSE in ClimAero is higher than that in NoAero during daytime (Fig. S6b-c). It is observed in Fig.S7a that, when averaging over Jan. 13th – Feb. 8th, the bias of 2-m temperature in ChemAero (0.48 °C) is lower than those in NoAero (0.79 °C) and ClimAero (0.52 °C). Comparing the absolute bias difference (°C) between ClimAero and NoAero (ClimAero-NoAero), and between ChemAero and NoAero (ChemAero and NoAero) in Fig. S7b, the ChemAero shows more improvement than ClimAero in the simulation of 2m temperature, particularly during the events of Jan. 15-19, and Feb. 3-9. In regards of wind speed at 10m, the overestimated wind speed in NoAero was decreased in ClimAero and ChemAero, with the averaged bias of 1.49 m s $^{-1}$, 1.45 m s $^{-1}$ and 1.44 m s $^{-1}$, respectively (Fig.S7c-d). Moreover, the RMSE in ChemAero was lower than that in ClimAero, particularly during 1700 LT to 0500 LT (Fig.S6e-f). The detailed day-to-day comparisons confirmed the significant temperature improvement by inclusion of WRF-Chem simulated hourly AOD fields during several events, including Jan. 16-19, Jan. 25, Jan. 28, Feb. 3-4, and Feb. 7-9.

Overall, the one-month results are statistically significant which indicated that the simulation with the inclusion of WRF-Chem simulated hourly AOD fields outperformed other two simulations and showed more improvement on the forecast of surface temperature and near surface wind speed than the simulation with climatological AOD fields. We will work on this issue and perform more detailed evaluations and analysis in the future, aiming to facilitate the future inclusion of aerosol-radiation interactions in our regional operational Numerical Weather Prediction system.

[Figure]

Fig. S6. Area-averaged (a) bias and (b) RMSE of simulated 2-m temperature (°C) in NoAero (blue), ClimAero (green) and ChemAero (red) over NCP area (defined in Fig. 1a), averaged from Jan. 13th – Feb. 8th 2017, and the mean improvement (%) of (c) RMSE in ClimAero (green) and ChemAero (red) relative to NoAero. (d-f) are same with (a-c), but for wind speed at 10m (m s⁻¹).

[Figure]

Fig. S7. (a) Temporal variations (00 UTC of Jan. 13th – 24 UTC of Feb. 8th, 2017) of area-averaged 2-m temperature bias (°C) simulated in NoAero (blue soild), ClimAero (green solid)

and ChemAero (red soild) over NCP area (defined in Fig. 1a); (b) same with (a), but for the difference of absolute bias (°C) between ClimAero and NoAero (ClimAero-NoAero, green bars), and between ChemAero and NoAero (ChemAero-NoAero, red bars). (c-d) are same with (a-b), but for wind speed at 10m (m s$^{-1}$).

**Dear ACP Editor:**

**We have addressed all the comments raised by both reviewers, and incorporated them in the revised manuscript. Please find below our itemized responses to the reviewer's comments.**

**Thank you very much for your consideration.**

**Sincerely,**
**Yang Yang, et al.**
* * *
**COMMENTS TO THE AUTHOR(S)**
**Reviewer 2**
The article is interesting and treats a topic of utmost relevance, that of aerosol impacts on Numerical Weather Prediction (NWP). The authors have analyzed in great detail a pollution case in Northern China during December 2-11, 2015 and examined the impact of including aerosol radiative forcing on several key meteorological variables. They found that aerosols have a large impact on shortwave radiative fluxes at the surface and consequently on 2m temperatures and wind speed using independent observations from various networks to establish that. These results are consistent with finding from other authors who highlighted the importance of a correct inclusion of aerosol fields particularly under extreme aerosol loads.

The paper deserves attention and with some refinements will be acceptable for publication. However, it is worthwhile to stress that case studies such as this may not be statistically significant, especially because extreme aerosol conditions were chosen. It would be necessary to run more cases, possibly entire seasons. I would encourage he authors to get in touch with the rest of the community and join an effort sponsored y WMO via various committees (WGNE, GAW and S2S) to run coordinate experimentation in regional and global models with the goal to gain a fuller picture of the aerosol pacts in NWP. Feel free to contact me directly about this.

**Response:**

Dear Angela,

We are really glad to be reviewed by you and get to know that different groups are working on this important topic. We will try to get in touch with you and the community in the near future and promote the operational application in our system; in this way, the long-term assessment of aerosol impacts over the northern China region would be possibly conducted routinely, not only confined on the scientific level.

We really appreciate your interest and insightful comments. To address this issue about the statistically significance of the aerosol induced impacts on weather forecast, we further conducted three sets of 24-hour forecasts for a longer period lasting 27 days (Jan. 13th- Feb. 8th, 2017), with no AOD field (NoAero), climatological AOD fields (ClimAero) and WRF-Chem simulated hourly AOD fields (ChemAero) included, respectively.

The results indicated that the temperature was underestimated (overestimated) during daytime (nighttime) in NoAero experiment. The temperature is reduced by the aerosol-radiation interactions by inclusion of either climatological or WRF-Chem simulated AOD fields (Fig. S1a), which tends to increase the bias during daytime, and decrease the bias during nighttime. However, the RMSE of temperature in ChemAero is lower than NoAero during the whole 24-hr forecast, particularly at 2000LT of nightfall with the reduction of RMSE reaching ~ 9%. While the RMSE in ClimAero is higher than that in NoAero during daytime (Fig. S1b-c). It is observed in Fig.S2a that, when averaging over Jan. 13th – Feb. 8th, the bias of 2-m temperature in ChemAero (0.48 °C) is lower than those in NoAero (0.79 °C) and ClimAero (0.52 °C). Comparing the absolute bias difference (°C) between ClimAero and NoAero (ClimAero-NoAero), and between ChemAero and NoAero (ChemAero and NoAero) in Fig. S2b, the

ChemAero shows more improvement than ClimAero in the simulation of 2m temperature, particularly during the events of Jan. 15-19, and Feb. 3-9. In regards of wind speed at 10m, the overestimated wind speed in NoAero was decreased in ClimAero and ChemAero, with the averaged bias of 1.49 m s $^{-1}$, 1.45 m s $^{-1}$ and 1.44 m s $^{-1}$, respectively (Fig.S2c-d). Moreover, the RMSE in ChemAero was lower than that in ClimAero, particularly during 1700 LT to 0500 LT (Fig.S1e-f). The detailed day-to-day comparisons confirmed the significant temperature improvement by inclusion of WRF-Chem simulated hourly AOD fields during several events, including Jan. 16-19, Jan. 25, Jan. 28, Feb. 3-4, and Feb. 7-9.

Overall, the one-month results are statistically significant which indicated that the simulation with the inclusion of WRF-Chem simulated hourly AOD fields outperformed other two simulations and showed more improvement on the forecast of surface temperature and near surface wind speed than the simulation with climatological AOD fields. We will work on this issue and perform more detailed evaluations and analysis in the future, aiming to facilitate the future inclusion of aerosol-radiation interactions in our regional operational Numerical Weather Prediction system.

[Figure]

Fig. S1. Area-averaged (a) bias and (b) RMSE of simulated 2-m temperature (°C) in NoAero (blue), ClimAero (green) and ChemAero (red) over NCP area (defined in Fig. 1a), averaged from Jan. 13[th] – Feb. 8[th] 2017, and the mean improvement (%) of (c) RMSE in ClimAero (green) and ChemAero (red) relative to NoAero. (d-f) are same with (a-c), but for wind speed at 10m (m s $^{-1}$).

[Figure]

Fig. S2. (a) Temporal variations (00 UTC of Jan. 13th – 24 UTC of Feb. 8th, 2017) of area-averaged 2-m temperature bias (°C) simulated in NoAero (blue soild), ClimAero (green solid) and ChemAero (red soild) over NCP area (defined in Fig. 1a); (b) same with (a), but for the difference of absolute bias (°C) between ClimAero and NoAero (ClimAero-NoAero, green bars), and between ChemAero and NoAero (ChemAero-NoAero, red bars). (c-d) are same with (a-b), but for wind speed at 10m (m s $^{-1}$).

**Minor comments and typos**

line 22, high-frequency
**Response:**
    Thanks, corrected.

line 66, episodic aerosol events
**Response:**

Thanks, corrected.

line 105, to facilitate the inclusion of…

**Response:**

Thanks, corrected.

line 116, was included

**Response:**

Thanks, corrected.

line 119 For these research studies using operational NWP systems, offline approaches were mostly used. Actually, in Remy et al 2015 and Mulcahy et al 2014 that was not the case and the interactive aerosols were run online.

**Response:**

Thanks for your comment. We have changed the sentence to "For these research serving for operational NWP systems, both online and offline approaches (that aerosol information were simulated by separate chemistry system and then offline coupled to NWP model) were widely used." In the revised manuscript.

line 143, in an NWP system

**Response:**

Thanks, corrected.

line 152, future applications

**Response:**

Thanks, corrected.

line 153, The remainder of the paper is organised: : : Please change all tenses in this paragraph to present.

**Response:**

Thanks, corrected.

line 168, National

**Response:**

Thanks, corrected.

line 169, Environmental - please re-run the paper through a spell and grammar checker to ensure that typos are corrected
**Response:**
   Thanks, corrected.

line 171, with a higher
**Response:**
   Thanks, corrected.

line 174 the Rapid Radiative
**Response:**
   Thanks, corrected.

line 181 The RRTMG
**Response:**
   Thanks, corrected.

line 185 was input
**Response:**
   Thanks, corrected.

line 186 integral
**Response:**
   Thanks, corrected.

lie 189 which was - please check that verbs are correctly conjugated
**Response:**
   Thanks, corrected.

line 190 the same configuration
**Response:**
   Thanks, corrected.

line 206 did you investigate the sensitivity of the model AOD to the choice of these ICs and BCs?

**Response:**

Thanks for the kind reminder! Actually for these heavily polluted region in winter, the initial and boundary conditions are really not so important as in the clean regions, since the pollutant accumulation are usually associated with the high-intensity emission emitted and unfavorable meteorology conditions. For boundary condition, the default profile in WRF-Chem model seemed Okay for this region. For initial conditions, several-days spin-up staring from clean case and going-on for 3-4 days accumulation is usually close enough to real case. We have tried MOZART boundary condition for summer and did see some differences. We may test the sensitivity of the modeled AOD to the choice of chemical ICs and BCs in the future.

line 216 were CARSNET (https://www.atmos-chem-phys.net/15/7619/2015/) observations available over the area? if yes, why were they not used?

**Response:**

Thanks for your comment. Currently, we don't think the CARSNET dataset is publicly released and we don't have official access to it neither, but we agree with you that the collaboration is helpful in research work. To address the importance of simulated AOD accuracy, we added the evaluations of modeled AOD and aerosol extinction coefficient against MODIS and CALIPSO satellite-based products, respectively.

The modeled AOD was evaluated against MODIS Terra and Aqua (Fig. S3). It was seen that WRF-Chem is capable to capture the AOD spatial distribution and also reproduced the transport paths during the event. The simulated high-valued AOD located in Henan on Dec. 6th, then the center moved to Hebei and Beijing on 7th and shifted to northeast areas afterwards. The variations of simulated AOD were in consistent with both Terra and Aqua with slightly overestimated peak value of AOD. In particular, the simulated shifting of AOD center to northeast areas was also observed in Aqua (Fig. S3r-s).

Fig. S4 displayed the vertical distribution of simulated 550nm aerosol extinction coefficient compared to those from CALIPSO. Four cross sections along CALIPSO paths on 6th to 9th December were shown. The results indicated that the model could generally reproduce the vertical distribution of extinction coefficients at 550nm in terms of comparable magnitude with those from CALIPSO, particularly on 6th, 7th and 9th, December. However, CALIPSO showed more high values at lower altitude (below 1km) that model failed to capture; the inconsistency may be associated with both CALIPSO retrieval uncertainties at the low altitude and the model itself.

We have added the Fig. S3-4 and the discussion in the revised manuscript (around L247- L255 and around L255-L264).

[Figure]

Fig. S3 The WRF-Chem simulated and MODIS observed spatial distribution of AOD on 6th-10th December (from left to right). The first (a-e) and third rows (k-o) are WRF-Chem simulations at 1000LT and 1300LT (MODIS path times) respectively. The second (f-j) and fourth (p-t) rows are MODIS Terra and Aqua observations, respectively. Gray areas in (f-j) and (p-t) denote the missing values.

[Figure]

Fig. S4 The WRF-Chem simulated 550nm AOD (shadings) on (a)1800UTC of 6th, (b) 0400UTC of 7th, (c)1700UTC of 8th, (d) 0400UTC of 9th December overlaid with CALIPSO paths (black thick solid). (e-l) denote the corresponding vertical distributions of aerosol extinction coefficient at 550nm from (e, g, i, k) CALIPSO and (f, h, j, l) model simulations. Gray areas in (e-l) denote the terrain.

line 237 / Figure 2 I think it would be good to have extra data from CARSNET if possible

**Response:**

Thanks for your insightful comment. We added the evaluations of modeled AOD and aerosol extinction coefficient against MODIS and CALIPSO satellite-based products, respectively. Please see more detailed discussion above.

line 244 most of them, do you mean the observations during the peak? See comment above.

**Response:**

Yes, the observation of AERONET are quite limit during the peak. We added the evaluations of AOD simulation with MODIS and CALIPSO satellite-based products. Please see more detailed discussion above.

line 245 were similar to

**Response:**

Thanks, corrected.

line 247 you need more observations to establish that

**Response:**

Thanks, we agree with you that the statement seems arbitrary without more observations here, we have deleted this sentence in the revised manuscript.

line 265 do you think this was because of the emission inventories used or the skill of the model or both? Please comment.

**Response:**

Thanks for your comment. From our experience, these biases in two directions are related with both the emission inventories used and the skill of the model, but more diagnostic should be conducted to gain solid conclusions. We have added the comment "It was note that there exits slight overestimation (underestimation) of the peak magnitude during 9th to 10th at Beijing and Shijiazhuang (Tianjin, Hebei and Henan);the overestimation in Beijing and Shijiazhuang is possibly associated with the frequent emission changes caused by emission-control-measures in reality which are not dynamically updated in the model; the underestimation is more related with the deficiency of model skills, such as missing heterogeneous reaction paths in the chemistry scheme." in the revised manuscript.

line 286 In the NoAero experiments were the aerosols completely missing from the simulation or was a climatology used?

**Response:**

NoAero experiments were the aerosols completely missing from the simulation, we have added the clarification about this issue as "The only difference between the two sets of forecasts is whether the aerosol radiative feedback is activated (Aero, with WRF-Chem simulated hourly AOD fields as input fields) or not (NoAero, no aerosol included), and other schemes remained the same." in the revised manuscript (around L214-L217).

line 302 if a climatology were used would this discrepancy be less severe? I am assuming that in the NoAero simulations there were really no aerosols.

**Response:**

Thanks for your comment. NoAero experiments were the aerosols completely missing from the simulation, we agree with you that the discrepancy of shortwave radiation would be less severe if a climatological AOD were used.

line 304 this type of bias in SW fluxes is huge

**Response:**

Thanks for your comment. The polluted episode is a severe event with the maximum of AOD exceeding 8 at Beijing. Therefore, the SW fluxes were profoundly overestimated due to the missing processes of strong forcing from aerosol-radiation interaction, in the NoAero experiment. Actually we suspect the aerosol-cloud interactions may play some role in reality as well.

line 321/Figure 6 At some stations the bias in SW fluxes is not improved as much as in Beijing - do you have an explanation for that?

**Response:**

Thanks for your comment. The magnitude of changes in SW radiation induced by aerosol-radiation interaction is associated with the magnitude of AOD. The AOD at

Beijing is much higher than those of Tianjin, Taiyuan and Jinan. Therefore, the biases in SW fluxes at these stations were not improved as much as that at Beijing. We have added the discussion about this issue in the revised manuscript (L351-L352).

line 341 are discussed

**Response:**

Thanks, corrected.

line 368 is this an average value? With the biases in SW radiation being so large I would have expected higher temperature biases.

**Response:**

Thanks for your comment. The temperature bias is the averaged bias over NCP domain and for the whole period during 6th to 10th December. We have clarified this issue in the revised manuscript.

line 420 / Figure 15, the wind profile at Beijing is quite different from observations in both Aero and NoAero experiments, do you have an explanation for that?

**Response:**

Thanks for your comment. The wind speed at lower layers is generally overestimated in our operational NWP system, but with much lower magnitude than those shown in Fig. 15. The large bias in Fig. 15 is probably related with the problem about the presentation of boundary layer processes for this period in the model.

line 450 very nice discussion of the impacts on the vertical stratification

**Response:**

Thanks for your comment.

line 461, please specify if an aerosol climatology was used in NoAero

**Response:**

Thanks, NoAero experiments were the aerosols completely missing from the simulation, we have clarified this issue in the revised manuscript (L495-L499).

line 520 the fact that aerosol-cloud interactions were not included in the study should be mentioned also at the beginning.

**Response:**

[revised manuscript text omitted]

---

## Author Response (AR2)

**Dear ACP Editor:**

**We have addressed all the comments raised by the reviewer, and incorporated them in the revised manuscript. Please find below our itemized responses to the reviewer's comments.**

**Thank you very much for your consideration.**

**Sincerely,**
**Yang Yang, et al.**
* * *
**COMMENTS TO THE AUTHOR(S)**

**Reviewer 1**

I commend the authors for their efforts in responding to my comments. The paper has significantly improved compared to the previous version. The new analysis with Climatological AOD is interesting. However, I have some minor issues related to this new analysis that need to be addressed before the paper can be accepted for publications.

**Response:**

We really appreciate your insightful and valuable comments. Please find below our itemized responses to your comments and concerns.

1) Could you please mention how the climatological AOD distribution is derived?

**Response:**

Thanks for your kind reminder. The climatological AOD at 550nm is set by the *aer_opt* option, and the constant value of 0.12 is used with a default fixed vertical profile. We have added the clarification about this issue in the abstract and summary of the revised manuscript.

2) The improvements in bias and RMSE (shown in Figure S6) between the ClimAero and ChemAero cases are normally within 1-3%. This means that the NWP agencies having limited computational capability can opt for a climatological AOD distribution rather than running a fully complex atmospheric chemistry model. I encourage the authors to quantify the performance gain in ChemAero relative to ClimAero and include that information in the abstract and summary of the paper.

**Response:**

Thanks for your comment and reminder. We added the discussion about this issue in the abstract (L53-L60) and summary (L565-L576) in the revised manuscript: "To verify the statistically significance of the results, we further conducted the 24-hour forecasts for a longer period lasting 27 days (Jan. 13$^{th}$ – Feb. 8$^{th}$, 2017), with no AOD field (NoAero) and WRF-Chem simulated hourly AOD fields (Aero) included, as well as constant AOD value of 0.12 (ClimAero), respectively. The results showed that the mean RMSE of 2-m temperature (wind speed at 10m) in Aero and ClimAero relative to NoAero was reduced by 3.95% (1.86%) and 1.23% (1.63%), respectively (not shown). The one-month results indicated that the simulation with the inclusion of WRF-Chem simulated hourly AOD fields outperformed other two simulations and showed more improvement on the meteorological forecast than the simulation with climatological AOD fields. More detailed evaluations and analysis will be addressed in a future article."

It's assumed that there are still some differences between the climatological AOD and real-time simulated AOD, in terms of the absolute values and vertical distributions, which would possibly lead to more significant thermodynamic changes vertically. In addition to parameters at surface, statistics of vertical parameters would be further conducted in future study.

Regarding the question whether the NWP agencies should afford expensive computational resource to include real aerosol field instead of climatological field, more studies on aerosol-radiation-interaction and also aerosol-cloud-interaction in different seasons should be conducted to answer this question. Recent study (Jiang et al., 2017) revealed potential influences of neglecting aerosol effects on the NCEP GFS precipitation forecast: the standard deviation of the forecast bias was significantly correlated with aerosol optical depth in Australia, the US, and China. Considering the complex and more localized aerosol-radiation-cloud-precipitation process in the model, compressive analysis and comparisons of real aerosol V.S. climatological aerosol effects on NWP should be conducted to draw a solid conclusion.

Ref.

Jiang, M., Feng, J., Li, Z., Sun, R., Hou, Y.-T., Zhu, Y., Wan, B., Guo, J., and Cribb,

M.: Potential influences of neglecting aerosol effects on the NCEP GFS precipitation forecast, Atmos. Chem. Phys., 17, 13967–13982, https://doi.org/10.5194/acp-17-13967-2017, 2017.

3) The bias increases in ClimAero and ChemAero between 0800 and 1100 local time (Figure 6) despite the fact that you are doing the right thing by allowing aerosols to affect the radiation. I understood that the model was already underestimating the daytime surface temperature and since aerosols always cool the surface, the bias got worse rather than improving during daytime. Can you please say which physical processes are responsible for switching of the model behavior between daytime and night time?

**Response:**

Thanks for your concern. We agree with you that the model performance is actually determined by the uncertainties from various aspects, the correcting of single aspect wouldn't work for all the cases; thus cold bias during daytime in our model system cannot be corrected by the inclusion of aerosol-radiation interactions, which is actually induced by other physical processes. According to our recent research, the overestimated soil moisture is partly responsible for the cold temperature bias during daytime, particularly over mountain areas (Zhong et al., 2020). The overestimated soil moisture in model, which turns to froze during winter, leads to overestimated soil thermal conductivity and thus different biases of surface-atmosphere process during daytime and nighttime: overestimated downward (from atmosphere to surface) ground heat flux and less sensible heat flux (from surface to atmosphere) during daytime, in contrast overestimated upward ground heat flux transported from soil to atmosphere at nighttime. Besides, other physical processes including the misrepresenting of snow cover in the model should be investigated and fixed in the future.

Ref.

Zhong, J, Lu, B., Wang, W, Huang, C, Yang, Y., 2020, Impact of soil moisture on winter 2-m temperature forecasts in northern China, *J. Hydrometeorology*, 21, 597-614.

[revised manuscript text omitted]

PM$_{2.5}$ (μg m$^{-3}$) at three major cities (a) Beijing (BJ), (b) Shijiazhuang (SJZ)

(c)Tianjin (TJ) and two provinces (d) Hebei (HB) and (e) Henan (HN).

[Figure]

Figure 8. (a–d) observed (black) and WRF simulated (NoAero: blue, Aero: red)

temporal variation of downward shortwave radaition at surface (W m$^{-2}$, right axis)

at (a) Beijing, (b) Tianjin, (c) Taiyuan and (d) Jinan, respectively. The grey areas indicate the simulated AOD (left axis) by WRF-Chem. (e–h) are same with (a–d), but for net radaition at surface (W m$^{-2}$).

[Figure]

Figure 9. (a–d) observed (black) and simulated (NoAero: blue, Aero: red) diurnal cycles of downward shortwave radaition at surface (W m$^{-2}$) averaged from 6[th] to 10[th]

Dec. 2015 at (a) Beijing, (b) Tianjin, (c) Taiyuan and (d) Jinan, respectively. (e–h)

are same with (a–d), but for net radaition at surface (W m$^{-2}$).

[Figure]

Figure 10. The differences (Aero minus NoAero) of (a) surface sensible heat flux and (b) surface latent heat flux (W m$^{-2}$, upward is positive) at 1300LT averaged from 6th to 10th Dec. 2015.

[Figure]

Figure 11. The bias of 2-m temperature (°C) at (a) 1100, (b) 1400, (c) 1700, (d) 2000 and (e) 2300 LT in NoAero averaged from 6th to 10th Dec.

2015, (f–j) are same with (a–e), but for Aero. The grey areas denote the areas of terrain height above 1000m.

[Figure]

Figure 12. Area-averaged (a) bias and (b) RMSE of simulated 2-m temperature (°C )

in NoAero (blue) and Aero (red) over NCP area (defined in Fig. 1a), averaged from

6[th] to 10[th] Dec. 2015, and the mean improvement (%) of (c) absolute value of bias and (d) RMSE in Aero relative to NoAero.

[Figure]

Figure 13. Daytime mean PBLH (m) in NoAero, (b) the difference between Aero and NoAero (Aero minus NoAero) and (c) the ratio of changes (%) averaged during

6[th] to 10[th] Dec. 2015.

[Figure]

Figure 14. NCP (defined in Fig. 1a) area-averaged vertical profiles of potential temperature (K, solid) and planetary boundary-layer height (m, dash) in NoAero (blue) and Aero (red) at 1400 LT of (a) 6[th], (b) 7[th], (c) 8[th], (d) 9[th] and (e) 10[th] Dec.

2015.

[Figure]

Figure 15. The 10m wind (vector) at 1100, 1400, 1700, 2000 and 2300 LT in (a–e)

NoAero and (f–j) Aero averaged during 6[th] to 10[th] Dec. 2015, shadings in (f–j) are simulated $PM_{2.5}$ concentrations ($\mu$g m$^{-3}$). (k–o) the difference of 10m wind (vector)

and wind speed (shadings) between Aero and NoAero (Aero minus NoAero).

[Figure]

Figure 16. The bias of 10m wind speed (m s$^{-1}$) at 1100, 1400, 1700, 2000 and 2300

LT for (a–e) overestimated sites and (f–j) underestimated sites in NoAero averaged during 6$^{th}$ to 10$^{th}$ Dec. 2015.    (k–o) the difference of absolute value of bias (m s$^{-1}$)

between Aero and NoAero (Aero minus NoAero). The grey areas denote the areas of terrain height above 1000m.

[Figure]

Figure 17. Same with Fig.12, but for wind speed at 10m (m s$^{-1}$).

[Figure]

Figure 18. (a–b) Observed (black) and simulated (NoAero: blue, Aero: red) vertical profiles of atmospheric wind speed (m s$^{-1}$) at (a) Bejing and (b)Xingtai at 0800LT

averaged from 6$^{th}$ to 10$^{th}$ Dec., (c–d) are same with (a–b), but at 2000LT.

[Figure]

Figure 19. The wind at 850hPa (vector) at 1100, 1400, 1700, 2000 and 2300 LT in

NoAero averaged during 6th to 10th Dec. 2015. The difference of (f–j) U and (k–o) V

wind speed between Aero and NoAero (Aero minus NoAero). The grey areas denote the areas of terrain height above 1000m.